# Correcting negatively-biased refractivity below ducts in GNSS radio occultation: An optimal estimation approach towards improving planetary boundary layer (PBL) characterization

Kuo-Nung Wang[1], Manuel de la Torre Juárez[1], Chi O. Ao[1], and Feiqin Xie[2]

[1]Jet Propulsion Laboratory, California Institute of Technology, 4800 Oak Grove Drive, Pasadena, CA 91109, U.S.A.
[2]Texas A & M University - Corpus Christi, 6300 Ocean Dr., Corpus Christi, TX 78412, U.S.A.

*Correspondence to:* Kuo-Nung Wang (Kuo-Nung.Wang@jpl.nasa.gov)

**Abstract.**

GNSS radio occultation (RO) measurements are promising in sensing the vertical structure of the Earth's planetary boundary layer (PBL). However, large refractivity changes near the top of PBL can cause ducting and lead to a negative bias in the retrieved refractivity within the PBL (below $\sim$2 km). To remove the bias, a reconstruction method with assumption of linear structure inside the ducting layer models has been proposed by Xie et al. (2006). While the negative bias can be reduced drastically as demonstrated in the simulation, the lack of high-quality surface refractivity constraint makes its application to real RO data difficult. In this paper, we use the widely available precipitable water (PW) satellite observation as the external constraint for the bias correction. A new framework is proposed to incorporate optimization into the RO reconstruction retrievals in the presence of ducting condition. The new method uses optimal estimation to select the best refractivity solution whose ~~precipitable water (PW )~~ PW and PBL height best match the ~~external PW measurements~~ externally retrieved PW and the known a-priori, respectively. The near coincident PW ~~measurements~~ retrievals from AMSR-E microwave radiometer instruments are used as an external observational constraint. This new reconstruction method is tested on both the simulated GNSS-RO profiles and the actual GNSS-RO data. Our results show that the proposed method can greatly reduce the negative refractivity bias when compared to traditional Abel inversion.

## 1 Introduction

The planetary boundary layer (PBL) is the lowest layer of the atmosphere ($\sim$2 km)~~that~~, and couples the surface to the free troposphere. Influenced mainly by surface friction, solar radiation, and turbulent transport of moisture, the PBL controls the energy distribution from the surface into the atmosphere. Through the turbulent winds along with cumuliform and stratiform clouds formation, the PBL can greatly affect the local weather as well as the global climate (Garratt, 1992). Due to its importance to the weather prediction community the PBL has been extensively studied with various sounding techniques for several decades.

Among the existing probing technologies, Global Navigation Satellite System (GNSS) radio occultation (RO), which provides high resolution atmospheric vertical profiles, has been used to characterize the PBL in recent years (von Engeln et al.,

2005; Sokolovskiy et al., 2007). GNSS RO is a limb sounding technique that precisely measures the GNSS signals phase delay received by low earth orbiting (LEO) satellites, through which the bending angle and accurate atmospheric refractivity profiles can be retrieved (Kursinski et al., 1997). As a remote sensing technique, GNSS RO acquire measurements over remote regions including marine areas where the PBL can be crucial for weather (Beljaars and Viterbo, 1998) and climate modeling (Zeng et al., 2004) and cannot be achieved by traditional radiosonde, tower, and field observations. A high vertical resolution of approximately 100 m (Gorbunov et al., 2004) is another advantage of GNSS RO over other passive remote sensing instruments (Curran, 1989). Additionally, the L-band GNSS RO signals can penetrate through clouds and precipitation (Solheim et al., 1999), which are common at the height of PBL top. These features make GNSS RO a valuable tool for sensing the PBL (Guo et al., 2011; Ao et al., 2012; Xie et al., 2012; Chan and Wood, 2013; Ho et al., 2015).

However, it is known that the large refractivity change associated with a strong inversion layer at the top of the PBL can cause severe negative biases in RO refractivity measurements ($N$-bias) (Sokolovskiy, 2003; Xie et al., 2006; Ao, 2007). The large temperature and moisture changes near the PBL top could lead to a sharp negative refractivity gradient such that the radius of curvature of the signal path becomes less than the radius of the Earth. This phenomenon, called ducting or super-refraction, occurs when dN/dr $\lesssim$ -157 (N-units/km) and can be frequently observed in the subtropics below 2 km. The layer where the ducting occurred is called the ducting layer. Due to the ~~transceiver~~ transmitter and receiver geometry of GNSS-RO, the tangent point of each ray path never locates within ducting layers, which theoretically can "trap" the signal whose tangent point is inside. As a result, the GNSS-RO bending angle measurements will loss the information inside the ducting layer~~will be missing in GNSS-RO bending angle measurements and~~, in which cannot be recovered ~~by~~ using solely GNSS-RO observations. The standard Abel inversion of the bending angle profile will always lead to a profile with no ducts and can cause a negative $N$-bias as large as 15% below the ducting layer ~~.~~ (Xie et al., 2010). Correcting the $N$-bias within the PBL is essential towards the use of RO in studying the vertical structure within the PBL. While the weather analyses can assimilate RO bending angles, which are unaffected by the refractivity bias caused by ducting, it is not clear that the analyses can optimally handle these high vertical resolution measurements. In addition, the analyses may be strongly affected by bias in the model, as evidenced by the low PBL height over the stratocumulus regions (Xie et al., 2012). Therefore, it is of great scientific interests to retrieve an unbiased PBL refractivity based on observations only.

To mitigate the $N$-bias and reconstruct refractivity profiles inside the boundary layer, a reconstruction method was proposed by Xie et al. (2006) hereinafter referred to as (Xie06). The paper confirmed that an infinite number of refractivity profiles are corresponding to one bending angle profile in the presence of ducting condition. A non-linear function used to describe the continuum of refractivity solutions was derived based on the Abel-retrieved refractivity profile. Choosing the correct parameter and profile from the continuum, however, depends on two assumptions that cannot be easily fulfilled. First, to use the surface refractivity constraint, the RO bending angle measurements are implicitly assumed to cover all altitudes and stop exactly at the Earth's surface. However, the real RO bending angle profiles often do not reach the surface due to a combination of receiver measurement errors and atmospheric variabilities (Ao et al., 2012). Second, the reconstruction method assumes that the top height of the ducting layer can be determined accurately. However, due to the high variability of the bending angle, identifying the impact parameter of the ducting layer in the real occultation could be challenging.

In this paper we present a new and improved reconstruction method that implements optimal estimation along with external measurements of precipitable water (PW) based on a modification of Xie06 approach. In section 2, the ducting effects and the reconstruction method in Xie06 are reviewed and our new approach using optimal estimation is described. The results for radiosonde simulation using the optimal estimation approach and the comparison with different PW observation sources are presented in section 3. In section 4 we validate actual GNSS RO data results using the proposed reconstruction method. The summaries and conclusions are provided in section 5.

## 2 Refractivity Reconstruction Method

### 2.1 $N$-Bias

~~For a single ray path~~Under the assumption of a spherically symmetric atmosphere, the impact parameter $a$ of a single ray path can be defined as:

$$a = rn(r)\sin\phi \tag{1}$$

where $n$ is the ~~refractivity~~ refraction index, $r$ is the distance from the center of curvature to each point of the ray path, and $\phi$ is the angle between the ray path and the radial vector. ~~Under the assumption of a spherically symmetric atmosphere, the~~ The accumulated bending angle of a GNSS-RO ray path can be calculated for a refractivity profile as (Fjeldbo et al., 1971):

$$\alpha(r_t) = -2n(r_t)r_t \int_{r_t}^{\infty} \frac{1}{n(r)} \frac{dn(r)}{dr} \frac{dr}{\sqrt{[n(r)r]^2 - [n(r_t)r_t]^2}} \tag{2}$$

where $\alpha$ is the bending angle and $r_t$ is the radius of the ray path at the tangent point. To retrieve the refractivity information from the bending angle measurement, equation (2) can be simplified by using the impact parameter defined as $a = n(r_t)r_t$ ($\phi = 90°$ in equation (1) at the tangent point) and assuming that the function $x = n(r)r$ is monotonically increasing with $r$:

$$\alpha(a) = -2a \int_{a}^{\infty} \frac{1}{n} \frac{dn}{dx} \frac{dx}{\sqrt{x^2 - a^2}} \tag{3}$$

so that the refractivity can be derived analytically as (Fjeldbo et al., 1971):

$$n(x) = \exp\left[\frac{1}{\pi} \int_{x}^{\infty} \frac{\alpha(a)\,da}{\sqrt{a^2 - x^2}}\right] \tag{4}$$

which is called the Abel inversion integral.

Abel inversion is extensively used in RO retrievals, based on the fact that in most cases the one-to-one relationship between the derived refractivity profile and the measured bending angle profile is valid. However, this relationship breaks down when ducting occurs (Sokolovskiy, 2003).

### 2.1.1 Ducting Effects

The refractivity in the neutral atmosphere is related to atmospheric temperature, pressure, and the water vapor pressure with the following equation (Smith and Weintraub, 1953):

$$N = 77.6\frac{p}{T} + 3.73 \times 10^5 \frac{e}{T^2} \tag{5}$$

where $N = (n-1) \times 10^6$ is the refractivity in N-units, $n$ is the refractive index, $p$ is pressure in mbars, $T$ is temperature in Kelvins, and $e$ is the water vapor pressure in mbars. Due to the large change of temperature and moisture, the refractivity

decreases rapidly across the PBL top as seen in Figure 1(a) between the height $h_m$ and $h_t$. The ducting condition occurs when the refractivity gradient exceeds the critical refraction, i.e., $dN/dr \lesssim -157$ (N-units/km), where $x(r) = n(r)r$ is no longer a monotonic function with respect to $r$. The height $h$ is defined (Sokolovskiy, 2003) as:

$$h = r - r_e \tag{6}$$

where $r_e$ is the radius of curvature at the Earth surface. As illustrated in Figure 1(b), the function $h(x)$, shown in black, can be

divided into four intervals, i.e., $h_1(x)$: the $x$ increases from the surface to the height $h_b$ to reach $x_b$; $h_2(x)$: $x$ further increases to $x_m$; from $h_b$ to $h_m$; while $h_3(x)$: is the ducting layer with refractivity gradient exceeding critical refraction, where $x$ decreases from $x_m$ at $h_m$ back to $x_b$ at $h_t$; and $h_4(x)$: $x$ increases again from $h_t$ and the monotonic relation between $x$ and $r$ is restored. Within the intervals of $h_2(x)$ and $h_3(x)$, the changing signs of the slopes results in non-monotonic relationship between $h$ and $x$.

For simplicity, here, we define the trapping layer, which includes the ducting layer and the layer underneath from $h_b$ to $h_t$ where the monotonic characteristic of $h(x)$ vanishes. Inside the trapping layer, the refractivity gradient is large enough to trap the signal with a tangent point between $h_b$ and $h_t$ and cause an infinite bending angle in its ray path. Because of the geometry of GNSS RO, in which both the transmitter and the receiver are located outside the Earth atmosphere, ~~all the signal with a tangent point~~ the tangent points of the received signals will not appear inside the trapping layer~~cannot be received~~. In other words, the

information between $h_b$ and $h_t$ is lost in the received signal, and the bending angle observation is not be able to cover this gap. ~~We can verify it~~ This gap can be noticed by examining equation (2). When ~~the tangent point lies~~ evaluating the bending angle with (2) inside the trapping layer, $h_b < r_t - r_e < h_t$, the term $n(r)r$ above the height $r_t$ becomes less than $n(r_t)r_t$ because of the negative gradient of $x$ between $h_m$ and $h_t$. ~~This will~~ It would lead to a negative value inside the square root ~~of~~ in equation (2) and ~~makes the numerical calculation of bending angle unphysical when multiple $x$ values exist~~ the solution is a complex

number for the bending angle inside the trapping layer which is unphysical. However, all the bending angle below $h_b$ can still be ~~retrieved~~ evaluated as the monotonic relationship of $h(x)$ is in place. ~~In other words~~ Namely, the rays with tangent points below the trapping layer can still penetrate through the trapping layer and arrive at the receiver, and so the bending angle of these rays can still be ~~retrieved through the regular Abel inversion~~ calculated by GNSS-RO.

Although the missing bending angle measurements in the trapping layer cause a gap in the $\alpha - r$ relationship, the $\alpha - a$ relationship, will remain seamless because the $x$ value at the top and the bottom of the trapping layer are identical (e.g., $x_b$ in Figure 1(b)). Therefore, we can still apply the standard Abel inversion upon the bending angle profile in the presence of the ducting. But since the $x(r)$ function is not monotonic within the trapping layer, the standard ~~Able~~ Abel inversion (4), which

assumes monotonic $x$, will lead to erroneous refractivity results below $h_t$. While the refractivity retrieval remains valid above $h_t$, the bending contribution inside the trapping layer is missing from the standard Abel inversion below $x_b$. Consequently, the Abel retrieved refractivity below $x_b$ will be negatively biased (negative N bias) (Sokolovskiy, 2003; Xie et al., 2006; Ao, 2007) which are shown as the grey curves in Figure 1(a) and 1(b). To correct the GNSS-RO refractivity retrieval bias in the presence of ducting layer inside the PBL, a bi-linear trapping layer model along with a reconstruction method were proposed by Xie06,

which is described in the next sub-section.

### 2.1.2    Bi-linear Trapping layer Model and the Reconstruction Method

Xie06 demonstrated that an infinite number of refractivity solutions can generate the same bending angle observation using equation (2). ~~Due to the information loss inside the trapping layer, the standard Abel refractivity retrieval introduces the most significant~~ Among all the refractivity solutions corresponding to the same bending angle profile, the one retrieved by

the standard Abel introduces the largest negative bias. To ~~simplify the problem~~ mathematically describe each solution, Xie06 assume that the $h(x)$ depicted in Figure (1b) can be approximated by a simple bi-linear model, e.g., two connected straight-line segments inside the trapping layer between $h_b$ and $h_t$:

$$h_2 \left( x \right) = h_b + \frac{h_m - h_b}{x_m - x_b} \left( x - x_b \right) \tag{7}$$

$$h_3 \left( x \right) = h_t - \frac{h_t - h_m}{x_m - x_b} \left( x - x_b \right) \tag{8}$$

Under this assumption, the missing information inside the trapping layer can be recovered by the parameterization of the $h(x)$ between $h_b$ and $h_t$, and the $h_1 \left( x \right)$ at the segment 1 can be derived analytically for given parameters $x_b$, $x_m$, $h_b$, and $h_t$:

$$h_1 \left( x \right) = h_A \left( x \right) + \frac{2}{\pi} \left( h_t - h_b \right) \left[ z - \left( 1 + z^2 \right) \tan^{-1} \left( 1/z \right) \right] \tag{9}$$

where $h_A \left( x \right)$ is the height function with respect to $x$ from the standard Abel refractivity retrieval (grey curve in Figure 1(b)),

and $z = \sqrt{x_b - x} / \sqrt{x_m - x_b}$. By using equation (9), all the possible refractivity profile solutions, or the continuum of the solutions, below $h_t$ can be produced given five parameters defining the trapping layer: $x_b$, $x_m$, $h_b$, $h_m$, and $h_t$.

To identify the best refractivity solution out of the continuum of the solutions, additional assumptions and constraints were proposed:

(A1) The height of the trapping layer top ($h_t$) is needed and can be derived from the standard Abel retrieved refractivity

profile. Due to the large bending angle (theoretically $\to \infty$) near both the top and bottom of the trapping layer, the peak in bending angle profile is assumed detectable and its corresponding impact parameter ($x_b$) can be identified accurately. The

height of the trapping layer top $h_t$, therefore, can be calculated in the Abel-inversion profile at the point with known $x_b$. However, accurate detection of the trapping layer top is challenging in practice as it will rely on the high resolution bending angle profile which could be very noisy without filtering. On the other hand, filtering process will reduce the vertical resolution, which leads to error in the parameter $x_b$. Therefore a more robust method to detect $x_b$ parameter is needed.

(A2) The standard Abel-retrieved refractivity profile near the top to trapping layer behaves like a square-root function. Expanding by Taylor series around $z = 0$ the Abel inversion result function $h_A(x)$ can be written as:

$$h_A(x) = h_t - \frac{4}{\pi}(h_t - h_b)z + (h_3 - h_2)z^2 + O(z^3) \tag{10}$$

Keeping the leading order term in $z$, the function $h_A(x) - h_t$ is assumed to behave like a parabola near $x_b$ and the derivative of its square can be written as:

$$\frac{d}{dx}[h_A(x) - h_t]^2\big|_{x \to x_b^-} = -C \tag{11}$$

so that $C$ can be determined through linear regression in $h_A(x)$ over 200 m below $x_b$ (Xie et al., 2006). By neglecting higher order terms of the derivative $C$ can be expressed as:

$$C \simeq \frac{16}{\pi^2} \frac{(h_t - h_b)^2}{x_m - x_b} \tag{12}$$

Since the parameters $x_b$ and $h_t$ are known, after $C$ is determined the parameter $x_m$ can be calculated given $h_b$ using equation

(12).

(A3) Based on global observations of high vertical resolution sounding over the ocean, the slope of $h_1(x)$ is assumed to be continuous at $h_b$ to the bottom of the trapping layer:

$$\frac{dh_1(x)}{dx}\bigg|_{x \to x_b^-} = \frac{dh_2(x)}{dx}\bigg|_{x \to x_b^+} \tag{13}$$

This assumption determines the parameter $h_m$ by linear extrapolation of $h_1(x)$ until $x = x_m$.

Up to this point, all five parameters are connected so that once the parameter $h_b$ is given, the other four can be determined. However, while a large number of $h_b$ can be used in a single GNSS-RO case to generate a family of candidate profiles, choosing the correct profile from the family is still challenging. To determine the parameter $h_b$ one needs an additional constraint:

(A4) The surface refractivity constraint is applied, which requires the RO bending angle observation extend to the Earth's surface, $x_0$,

$$h_1(x_0) = 0 \tag{14}$$

i.e. from the whole family of refractivity profiles only the one with minimum height starting at zero will be chosen as the best solution of the reconstructed profile. However, for a number of reasons, including measurement noise and tracking error, horizontal variability, and diffraction effects, the retrieved profiles do not often reach the Earth's surface (Ao et al., 2012). Thus, the surface constraint is very difficult to be fulfilled and becomes the main obstacle in applying the reconstruction method in
practice.

As shown above, the reconstruction method in Xie06 depends upon several conditions which may not be achievable. In this paper, we aim to refine this reconstruction method by improving the applicability of A1 and replacing the surface refractivity constraint in A4 with the combination of optimal estimation along with the external PW observation constraint. Furthermore, an improved method to determine the trapping layer top ($x_b$) is proposed and the bi-linear parametrization model of the trapping
layer is modified to reflect smoother and more realistic structure of the $h(x)$ curve.

## 2.2   Optimal Estimation Implementation

Our approach retains equation (9) from Xie06 as the core concept of the retrieval method. First, to improve the determination of the height of ducting layer, a refined algorithm is developed, which no longer determine $x_b$ only by the noisy bending angle profile but also through optimal estimation iteration. Secondly, the ~~PW measurement~~ retrieved PW from ancillary data is used
to replace the surface constraint in A4 (Xie06). This research focuses on correcting the $N$-bias below the trapping layer and derive the key parameters such as duct altitude, thickness, and refractivity gradient information.

To implement these two major changes to the reconstruction algorithm, the basic parametrization process is required to be modified as described in the following subsection.

### 2.2.1   Parameterization

In Xie06 reconstruction method, all five parameters are connected and only one free parameter, $h_b$, is needed to define each refractivity profile. To relax the $x_b$ determination constraint in A1 we use a different approach.

First, the highest 100 m of $h_1(x)$ will be replaced by the linear extrapolation from below, ~~whose~~ its slope is determined by linear regression ~~applied from~~ between 100 m ~~to~~ and 200 m below the height ~~$x_b$~~$h_b$. The reason is that the top of the analytical solution calculated by equation (9) is ~~very~~ sensitive to the mis-modeling of $h_2(x)$ and $h_3(x)$, which are assumed to be bi-
linear segments between $h_b$ and $h_t$. ~~Due to the characteristics of the Abel inversion, the impact of ducting on the GNSS-RO observation will be most significant at the height ~ $h_b$, and spurious~~ Spurious spikes and fluctuations ~~could show up~~ are found at the top 100 m of the function $h_1(x)$ if the straight line assumption ~~is violated~~ inside the critical layer is violated by the data or the parameters, e.g. $h_b$, ~~were~~ are not appropriately estimated from Equation (12). ~~These fluctuations can cause unphysical refractivity profiles where multiple refractivity values coexist at the same altitude and making the continuity of~~ While these
fluctuations in $h_1(x)$ are usually small (<20 m) and typically occurred within 100 m below $h_b$, they can significantly change the slope at $h_b$, which makes the $h_2(x)$ value obtained from assumption (A3) ~~very difficult to apply. Fortunately, the range of the fluctuation is typically not large so that we can simply replace it~~ inconsistent with having one refractivity for each impact parameter below $h_b$. Therefore, we replaced the top portion of $h_1(x)$ with a linear extension of ~~$h_1(x)$ below~~ the curve below to

~~remove the fluctuation~~. Note that the variable $h_b$ changes when the ~~highest 100 m~~ top of $h_1(x)$ ~~below $x_b$~~ is replaced. To avoid the inconsistency between the ~~changed~~ new $h_b$ and the ~~original~~ originally specified $h_b$, $x_m$ is chosen over $h_b$ as the ~~new~~ "free" variable to ~~define a profile .~~ construct a profile inside and below the critical layer

Second, instead of assuming $x_b$ as known, both parameters $x_b$ and $x_m$ will be used as "free" variables as is $h_b$ in the Xie06

approach. Namely, a pair of $x_b$ and $x_m$ are required to determine one refractivity profile in this new parametrization model. With both modifications, A3 from section 2.1.2 can be easily applied to find $h_b$ and $h_m$ by extending the top of $h_1(x)$ with the same slope until $x = x_b$ and $x_m$, respectively (see Figure 1b). These modifications, obviously, lessen the dependence upon A1 but put more reliance on the identification of multiple parameters $x_b$ and $x_m$, which requires other constraints to assist choosing the best candidate. In the next subsection we will investigate the using of PW observation as additional constraint to

choose the correct profile among the solutions.

### 2.2.2 Precipitable Water as the External Constraint

Precipitable water, PW, is the total column water vapor content in the Earth atmosphere. ~~Since~~ In this research PW is chosen as the constraint over other physical quantities for several reasons. First, most of the water vapor in the atmosphere is located within the PBL ~~,~~ so that accurate PW observations can provide extra information below the ducting layer to assist GNSS-RO

retrievals. As an example shown in Figure 2, each refractivity profile candidate corresponding to the same bending angle shows distinctive PW values. This is reasonable since the refractivity is strongly related to the water vapor content in equation 5 and the larger PW corresponds to greater refractivity. ~~As a result, PW is a suitable option to constrain the refractivity solution.~~

~~PW measurements are widely~~ Second, PW are available globally from a variety of sensors (Millan et al., 2016). The microwave radiometry remote sensing technology is used on board several Earth observing satellites including AMSR-E (Kawan-

20 ishi et al., 2003), AMSR-2 (Imaoka et al., 2010), TMI (Kummerow et al., 1998), and SSM/I (Alishouse et al., 1990) to acquire accurate PW ~~measurements~~ observations over oceans. Also, the laser-diode sensor on board airborne platform such as AMDAR (Petersen et al., 2016) and ground based GNSS receiver networks (Yuan et al., 2014) can provide PW observations over land. Here we focus on the AMSR-E observations on board NASA's Aqua satellite, while PW observations from other sources can be applied to this method as well. AMSR-E conically scanned the Earth surface with water vapor band ($\sim$22 GHz) and provided

the precipitable water estimates over the oceans with limited error ($\sim$0.6 mm) (Wentz and Meissner, 2000). While the PW calculated by GNSS-RO is negatively biased in the presence of ducting condition, the collocated AMSR-E PW ~~measurement~~ can be used to select the most optimal solution among the candidate profiles.

~~To compare the retrieved refractivity profile with the PW observations, a forward model to calculate PW from~~ Third, PW value of a candidate refractivity profile ~~was constructed~~ can be easily calculated to compare with the PW observations. In this

study, the iterative direct method (Kursinski and Hajj, 2001) utilizing the temperature information from European Centre for Medium-Range Weather Forecasts (ECMWF) atmospheric analysis is used to derive the PW from each candidate refractivity profile. The high resolution ECMWF analysis data (TL799L91) used in this research have 91 vertical levels from the surface to 0.01 hPa and 0.25° horizontal resolution. The data is modeled at every 6 hours and unevenly sampled in vertical space which has higher resolution near surface ($\sim$40 m). The core concept of the direct method is derived by combining the hydrostatic and

ideal gas laws. It can be shown that the relation of the atmospheric pressure at different levels:

$$p_{i+1} = p_i \left( \frac{T_i}{T_{i+1}} \right)^{\bar{m}_i \bar{g}_i / R(dT/dz)} \tag{15}$$

where $p$ is the pressure, $T$ is the temperature, $i$ is the index of each height interval, ~~$\bar{m}$ is the mean molecular mass,~~ $\bar{g}$ is the mean gravitational acceleration, $R$ is the universal gas constant, and $z$ is height. ~~Using this equation along with~~ $\bar{m}$ is the mean molecular mass of atmosphere which takes both dry air and vapor into account:

$$\bar{m}_i = m_d \frac{p_i - e_i}{p_i} + m_v \frac{e_i}{p_i} \tag{16}$$

where $m_v$ and $m_d$ are the molecular mass of dry air ($\sim 28.97$ g/mol) and water vapor ($\sim 18.02$ g/mol), respectively. Using the equation (15) along with the refractivity equation (5) one can solve the water vapor pressure profile $e$ iteratively by updating $\bar{m}$ at each step and the convergence at each height interval can be reached in one or two iterations.

The calculated water vapor profile, however, may not reach the surface of the Earth in most of the cases while the PW calculation requires the information of water content in the atmosphere all the way to the surface. Hence, a reasonable assumption, which can be observed in many cases, is made that the moisture within the boundary layer is well-mixed and the specific humidity $q$ is constant from the surface up until the lowest point of the RO retrieved profile. The specific humidity profile $q(h)$ can be simply calculated (Stull, 2015) by :

$$q = 622 \times \frac{e}{p} \tag{17}$$

By assuming the value of $q$ at the surface equals to the $q(h)$ at the lowest height, we can calculate the precipitable water with the integration (Millan et al., 2016) from surface to the height when temperature reaches $230K$ (Kursinski and Hajj, 2001):

$$PW = \frac{1}{g} \int_{P_0}^{P_\infty} q(p) \, dp \tag{18}$$

where the pressure profile $p$ can be extrapolated from the lowest profile height to the surface using an exponential fit function.

### 2.2.3  Optimal Estimation

To choose the best refractivity profile from a family of candidates with different $x_b$ and $x_m$, an optimal estimation method (Rodgers, 2000) is used based on a bayesian solution that minimizes the cost function of a linear inverse problem. In this method, the state vector $\boldsymbol{s}$ consists of two variables,

$$\boldsymbol{s} = \begin{bmatrix} x_b \\ x_m - x_b \end{bmatrix} \tag{19}$$

that can be connected to the observation vector $\boldsymbol{y}$ with a forward model $\mathbf{F}$ where

$$\boldsymbol{y} = \mathbf{F}(\boldsymbol{s}) \tag{20}$$

Because $x_b$ and $x_m$ are related, the second component of the state vector $\boldsymbol{s}$ is set as $x_m - x_b$ instead of $x_m$ so that the two components can be treated independently. The ~~PW measurement~~ observed PW is the observation vector $\boldsymbol{y}$ for the optimal estimation problem:

$$\boldsymbol{y} = \begin{bmatrix} PW \end{bmatrix} \tag{21}$$

The forward model $\mathbf{F}$ to calculate the measurement $\boldsymbol{y}$ for each given state $\boldsymbol{s}$ is described in sub-sections 2.2.1 and 2.2.2. The Jacobian matrix $\mathbf{K}$ defined as:

$$\mathbf{K}_n \equiv \left. \frac{\partial \mathbf{F}(\boldsymbol{s})}{\partial \boldsymbol{s}} \right|_{\boldsymbol{s}=\boldsymbol{s}_n} \tag{22}$$

is calculated numerically from the variation of $\mathbf{y}$ after perturbing the corresponding state $\mathbf{s}$ at the iteration step $n$. Defining $\mathbf{C}_{\boldsymbol{s}_0}$ and $\mathbf{C}_{\boldsymbol{y}}$ as the error covariance matrices of the a priori state $\boldsymbol{s}$ and the measurement $\boldsymbol{y}$, one can estimate the best solution
of $\boldsymbol{s}$ iteratively:

$$\hat{\boldsymbol{s}}_{n+1} = \boldsymbol{s}_0 + \left( \mathbf{C}_{\boldsymbol{s}_0}^{-1} + \mathbf{K}_n^T \mathbf{C}_{\boldsymbol{y}}^{-1} \mathbf{K}_n \right)^{-1} \mathbf{K}_n^T \mathbf{C}_{\boldsymbol{y}}^{-1} \left[ (\boldsymbol{y} - \boldsymbol{y}_n) - \mathbf{K}_n (\boldsymbol{s}_0 - \hat{\boldsymbol{s}}_n) \right] \tag{23}$$

where $\boldsymbol{s}_0$ is the a-priori guess of the state $\boldsymbol{s}$ and superscript $T$ denotes the transpose of the matrix. For this study, the state a-priori $x_b$ is determined by the impact parameter where a sharp transition occurs in the bending angle profile. The determination process using the step function correlation is described in Appendix A. However, the a-priori information of the parameter
$x_m - x_b$, which is highly correlated with the "strength" of ducting, cannot be obtained directly from the current GNSS-RO or AMSR-E measurement. In this study the $x_m - x_b$ a-priori value is chosen as the constant of 250 m, which is approximately the average number of $x_m - x_b$ from all the radiosonde profiles (19 cases) used in this study.

To calculate the covariance matrix, the uncertainty of each variable is required. The uncertainty of $x_b$ is set as $\pm 40m$ mentioned in Appendix A and used to form the $\mathbf{C}_{\boldsymbol{s}_0}$ matrix. The uncertainty of $x_m - x_b$, on the other hand, is not known and
the constant a-priori we chose is not based on any reliable sources. While the standard deviation of $x_m - x_b$ is $\sim 80$ m in the radiosonde profiles, we conservatively set the uncertainty of $x_m - x_b$ as large as $\pm 400m$ to allow the parameter for large flexibility and insensitivity to the a-priori we chose. The AMSR-E PW ~~measurement contains the~~ retrieval contains an error of $\sim 0.6$ mm, but ~~considering collocation errors~~ additional errors could rise from RO - AMSR-E collocation distances and forward modeling. Therefore, the conservative PW margin of 1 mm is used as the uncertainty of the PW observation in the
$\mathbf{C}_{\boldsymbol{y}}$ matrix. Both $\mathbf{C}_{\boldsymbol{s}_0}$ and $\mathbf{C}_{\boldsymbol{y}}$ are generated as simple diagonal matrices. Given appropriate initial conditions for all the least square fit included, the iterative process of equation (23) normally converges in a few iterations. The estimation results select the refractivity profile best fitted to the given $x_b$, $x_m - x_b$ apriori and PW observations, and correct the $N$-bias and provide the PBL top information including its altitude and refractivity gradient. It should be emphasized that the optimal estimation also creates a framework for solving the ill-posed inversion problem of refractivity retrievals under ducting by incorporating

multiple external constraints. In addition to PW constraint, the flexibility of the optimal estimation framework allows to use other physical constraints to correct the $N$-bias in the presence of ducting.

## 3   Simulation Results

To test and validate our algorithm, we conduct a simulation study utilizing radiosonde measurements from the VAMOS Ocean-Cloud-Atmosphere-Land Study (VOCALS) campaign (Wood et al., 2011). The VOCALS campaign dataset is used because of its location at the Southeast Pacific Ocean which has the world's most persistent subtropical stratocumulus deck at the top of the boundary layer (Bretherton et al., 2004). Moreover, the region also has one of the highest frequency of ducting condition (Lopez, 2009) and lead to large N-bias in RO refractivity retrievals (Xie et al., 2010). 19 observations with strong refractivity gradient at the top of PBL are selected for the simulation, and 6 among them are collocated with AMSR-E measurements (Figure 3 and Table 1). The x-h curves of these 6 cases are shown in Figure 4. Original RAOB $h(x)$ profiles are shown as the light red lines in Figure 4, which are non-monotonic functions in the trapping layer near ∼1.5 km for all six cases. Using the RAOB refractivity profiles as reference, we generate an observed bending angle which is then Abel-inverted to simulate the standard retrieved GNSS-RO refractivity profiles. While $x$ is not monotonically increasing in the RAOB refractivity profiles, the forward calculation of equation (2) should be used in here to generate the RO bending angle. Note that the potential errors caused by horizontal refractivity gradient are neglected in the bending angle simulation. The resulting standard Abel retrieval (x-h curves) are shown as black dotted lines. The Abel retrieval diverge from the RAOB profiles beneath the top of ducting layer and cause negative bias in the $x$ profiles below. The corresponding refractivity profiles for these 6 cases are shown in Figure 5, where the standard Abel-retrieved RO refractivity profiles (dotted) contain large negative biases below the trapping layer when compared to the original RAOB profiles (light red lines). The collocated ECMWF analysis profiles are also shown as light green lines in both figures. The ECMWF analysis tends to underestimate the ducting layer height and the refractivity gradient inside, which causes negative refractivity biases at lower altitude when compared to the radiosonde measurements. Since VOCALS results were not assimilated in ECMWF analysis, these two data sources can be regarded as independent. The statistically low PBL heights in ECMWF, which were extensively observed in the region, implies an erroneous refractivity profile below the ducting layer. This difference has been attributed to the model physics and assimilation process limitations (Xie et al., 2012). Even though ECMWF and other NWP system assimilate both GNSS-RO bending angles and AMSR-E radiances, it is not clear that the full vertical resolution of the measurements can be taken into account. Thus an independent, unbiased, refractivity retrieval outside of NWP data assimilation systems remains extremely valuable.

The biased GPS-RO refractivity simulations are then processed with the proposed optimal estimation method and compared to the original radiosonde refractivity profiles. The PW values calculated by equation (18) for each case from the RAOB profiles are shown in the lower-left corner of each panel along with collocated AMSR-E measurements. To validate our reconstruction method in this section, the background temperature and pressure profiles required in the direct method are given by radiosonde observations which are regarded as truth. The reconstruction results using the RAOB PW are presented in both figures as dark red dashed lines. As Figure 5 shows, when the unbiased RAOB PW are used the reconstructed profiles can effectively reduce

the $N$-bias and provide accurate estimates below the trapping layer. Optimal estimation also provides the estimated parameters $x_b$ and $x_m$, which can be used to identify the altitude, thickness, and the refractivity gradient of the ducting layer. A discrepancy can be observed at the straight line section in the trapping layer corresponding to $h_2(x)$ and $h_3(x)$, where the original RAOB refractivity profile is not represented by two straight lines as we assumed. Fortunately, this approximation only induces small difference and has little impact on the reconstructed profile below the trapping layer. However, the $h_3(x)$ function inside the ducting layer is sensitive to the $x_b$ location and the slope at the top of the $h_1(x)$. This could lead to large refractivity differences from the true profile when the $x_b$ is not accurately determined or the slope of $h_1(x)$ and $h_2(x)$ near $x_b$ are not continuous as expected. This error cannot be corrected without adding additional constraints or measurements to further determine the vertical structure inside the trapping layer.

In practice, the PW information from RAOB is not always available for nearby GNSS RO soundings due to the sparsity of radiosonde stations in remote areas. To demonstrate the ability of using other ancillary PW sources in the proposed algorithm, the reconstructed profiles using the collocated ECMWF ~~PW~~ and AMSR-E PW ~~measurements~~ are also presented in Figure 5 as dark green dashed lines and blue dashed lines, respectively. All 3 different PW values used in the reconstruction method are listed in the lower-left corner of each panel. The PW value acquired from the three external sources (RAOB, ECMWF, AMSR-E) in all 6 cases are greater than the ones calculated from the negatively biased Abel-inverted profiles, which suggests dry biases in the Abel retrievals inside the boundary layer when ducting occurs. Therefore, the reconstructed profiles from the optimal estimation with larger external PW should lead to larger refractivity inside PBL and mitigate the $N$-bias.

The statistical results of total 19 RAOB cases using the reconstruction method with the radiosonde PW are shown in Figure 6(a). The refractivity difference is defined as:

$$\frac{N_{RO} - N_{RAOB}}{N_{RAOB}} \times 100\% \tag{24}$$

where $N_{RAOB}$ is the radiosonde refractivity, and $N_{RO}$ is the standard Abel refractivity retrievals in dotted lines or the reconstructed profiles in red lines. As depicted in Ao (2007), the negative $N$-bias reaches the greatest value (-8% to -17%) near the height of $h_t$ and decreases to $\sim$-5% near the Earth surface. On the other hand, using the optimal estimation method the retrieved refractivity profiles remain unbiased (<1%) below the trapping layer. The large error up to +13% at the top of the ducting layer ($h_3(x)$) is due to presence of the sharp refractivity gradient and minor $x_b$ and $h_m$ estimation differences from the RAOB profiles could lead to large difference. While the error mostly occurs around the top of the ducting layer, they have very limited effects to the estimated profile below and the character inside the trapping layer.

However, the errors in external PW constraints will affect the reconstruction results. As presented in the Figure 6(b), while the reconstructed results using ECMWF PW reduce the $N$-bias, it still leads to a small negative bias in the reconstructed result (-1.54% in average) than the ones reconstructed from the RAOB PW (-0.01% in average). This may be due to a systematic underestimation of PW by the ECMWF analysis. Approximately, 1 mm of the PW bias can cause $\sim 3\%$ refractivity bias at the height $h_t$ and $\sim 1\%$ at the surface. Although the slight negative bias caused by lower PW values ($\sim 1mm$) could reduce its reliability, these results suggest that ECMWF analysis can still be used to improve the retrieval under the trapping layer.

The statistical results of the 6 cases using the 3 different PW sources are shown in Figure 7. While the reconstruction retrievals using ECMWF PW is negatively biased below the trapping layer, the results using the collocated AMSR-E measurements tend to be non-biased in general because of relatively better agreement between AMSR-E and radiosonde PW~~measurements~~. A positively biased outlier in AMSR-E PW reconstruction results can be identified as case 5, whose AMSR-E PW is apparently larger than those from other sources. The AMSR-E, ECMWF, and RAOB PW value comparisons are presented with the scatter plot in Figure 8, where ECMWF PW shows a clear negative bias compared to the RAOB PW. The large difference of PW in case 5 and case 1 may be due to the large distance (431 km) and the temporal difference (1.5 hr) between the AMSR-E measurement and the RAOB location, respectively. However, the cause of the PW difference in case 4 is unknown. While more analysis is needed to identify the true cause of the limited bias on each individual case, the microwave PW ~~measurement~~ retrieval is still proven to be another useful constraint when closely collocated with GNSS-RO and provide a feasible solution for GNSS-RO during ducting.

## 4 Actual GNSS-RO Data Results

We now apply our reconstruction method on actual COSMIC RO data. Eight COSMIC occultations collocated with VOCALS radiosondes (Figure 9) are chosen. ~~Four~~ Three criteria are utilized for choosing these cases: a spatial distance of less than 300 km, a temporal difference of less than 3 hours, the lowest height of the GPS-RO refractivity profile reaches below 1 km to ensure the trapping layer is included~~, and no double or complex~~. We also exclude the cases with complex x-h structure inside the trapping layer which ~~is beyond the scope of this research~~can heavily violate the bilinear assumption, and the cases with multiple ducting layers which makes the equation (9) inapplicable. Approximately 15% of the total number of cases are ruled out by these two additional requirements. The first 3 GPS-RO cases, have both collocated radiosondes and AMSR-E measurements and were numbered as cases 1 to 3 in Table 1. The other five cases which do not share the collocated RAOB measurements are numbered 7 to 11 for clarity. In practice, RAOB temperature and pressure profiles at the GNSS-RO collocation may not be available for the RO's PW calculation when using the direct method described in Section 2.2.2. Therefore, in this section we also include the ECMWF analysis profiles to compute PW for the GNSS-RO and compare them with corresponding radiosonde and AMSR-E measurements.

The results are shown as dashed lines in Figure 10. Similar to the simulation results in the previous section, the actual COSMIC RO refractivity profiles (dotted lines) are negatively biased compared to the collocated RAOB and ECMWF analysis. This negative refractivity bias leads to smaller GPS-RO PW than the one calculated from given RAOB profiles, ECMWF profiles, and AMSR-E measurements. The reconstructed results correct the bias to different degrees based on the source of PW ancillary data. Two main differences can be observed when comparing the reconstructed profiles to the reference collocated radiosonde profiles. First, the refractivity profile above $h_t$ from RO and radiosonde is not exactly the same, against the assumption that the bias only comes from super-refraction. This can bias the PW value calculation for all possible candidate profiles. Second, the estimated $x_b$ in reconstruction results can have at most 200 m difference with the corresponding radiosonde observations and cause the different shape of the reconstruction even if the PW is obtained accurately. These two differences can be caused

by the spatial/temporal difference between RO and RAOB observations, which are normally more than 200 km and 1 hour apart. For example, the reconstructed profile for case 3, COSMIC sounding is only 10.7 km and 1.18 hr apart from the RAOB location, agrees well with the RAOB profile in $x_b$ and the refractivity profile above. Another possible cause of $x_b$ discrepancy is the error in GNSS-RO measurement due to horizontal inhomogeneity in the atmosphere and the ionosphere (Zeng et al., 2016)

. In ducting conditions, this error can be amplified and shift the impact parameter of boundary layer top for more than 100 m. While addressing the horizontal inhomogeneity is beyond the scope of this article, the impact of horizontal refractivity gradient on the reconstruction method can be further investigated in future work.

The statistical results of the refractivity difference compared to the collocated radiosondes are shown in Figure 11. Two main differences mentioned in the previous paragraph can be easily seen: the differences above the ducting level can be as

much as 5%, and the $x_b$ difference cause $< -10\%$ error above $h_t$ and $> 10\%$ error under $h_t$. The not-so-closely collocated RAOB profile PW (red lines) can still maintain the $N$-bias below the trapping layer less than 5% in all the cases without bias. The 3 cases using AMSR-E PW ~~measurements~~ retrievals shown in blue lines agree better with the reference RAOB profiles, which can be attributed to the unbiased AMSR-E PW observation. Like the simulation results, the reconstructed profiles using ECMWF PW are negatively-biased ($\sim -2\%$) against the ones using other PW sources. This is because of the PW calculated

by ECMWF is negatively biased (Figure 8). Overall, our results show that reconstructed profiles utilizing external PW sources can substantially reduce the negative $N$-biases and limit the error to within 5% with zero mean below $h_t$ from the 15% negative error in the standard-Abel refractivity retrievals.

## 5   Conclusions

GNSS-RO has been extensively used in atmospheric profiling and weather forecasting. But the RO profiling in the presence of

20 the ducting layer remains a challenge. Ducting, which occurs when the refractivity gradient is large enough to trap the GNSS signals inside the Earth atmosphere, can cause ~~information loss inside~~ the lack of bending angle information in the trapping layer . As the ~~bending angle measurements lose~~ retrieved bending angle loses its one-to-one relationship with the ~~refractivity profiles~~atmospheric refractivity, the standard Abel inversion will give the refractivity solution with the largest negative bias. By approximating an analytical solution to the profile below the trapping layer and introducing a series of constraints, the method

by Xie06 is able to reconstruct the profile based on GNSS-RO observations. However, the reconstruction method is based on several idealizations difficult to be achieved from real measurements. To develop a practical reconstruction method, this paper validated a new implementation framework to incorporate constrained optimal estimation into ducting layer RO retrievals.

The proposed method modified the parametrization process to include more free parameters and reduce the reliance on idealized assumptions. The optimal estimation method is used to select the candidate that minimizes the cost function, which is

30 defined by the difference between the known reference (i.e. ancillary PW observations and state a-priori) and those calculated from each retrieval. PW ~~measurements~~observations, which can be obtained by remote sensing instruments such as AMSR-E, can serve as an external constraint in the reconstruction method. The process to infer the boundary layer height from bending angle profiles has also been refined to provide a robust and accurate estimation of a-priori ($x_b$). The new reconstruction method

has been applied to both the simulated GNSS-RO profiles and actual GNSS-RO data. The results show that given accurate PW, the proposed method greatly reduces the $N$-bias to less than 1% in simulation and 5% in actual cases. While the method cannot fully reconstruct the vertical structure inside the trapping layer, the iterated parameters are able to give improved estimation of PBL top features including ducting layer altitude, thickness, and the refractivity gradient.

To improve this reconstruction technique, several sources of uncertainty need to be further examined. The bias in different PW sources should be identified before being used as constraints, and the chosen PW ~~measurements~~ observations should be located close enough to GNSS-RO spatially and temporally. Currently, ECMWF analyses available at all locations have also been applied and shown to produce a bias $< 5\%$. Also, the assumptions of constant specific humidity from surface up to the minimum height of RO sounding and discontinuity of $x(h)$ slope at the bottom of trapping layer may cause errors. The optimal
estimation method developed in this work can be improved by incorporating other potential observations and constraints in the future and will help better characterize the vertical structure of the PBL globally using GNSS-RO measurements.

## Appendix A: Determination of $x_b$ in the Bending Angle Profiles

In this appendix we describe a new method to detect the impact parameter $x_b$ where the ducting occurs. Theoretically, the bending angle should reach infinity when the tangent point of the signal path is located inside the trapping layer (Sokolovskiy,
2003). In practice, the infinite value of bending angle is not observable in a finite observation, but the singularity in bending angle close to the impact parameter $x_b$ will result in a sharp transition. Where the sharp transition of the bending angle measurement is located can provide us valuable information on an a-priori $x_b$. In this paper, the parameter $x_b$ is determined by the peak of the correlation between the high resolution bending angle profile and a step function using a two-step approach, which is similar to the wavelet covariance transform (WCT) method proposed by Ratnam et al. (2010). The step function we
used is +1 of its lower 500 meters and -1 of its higher 500 meters which is similar to the shape of the bending angle profile affected by ducting that the transition from -1 to +1 in the step function matches the sharp transition of the bending angle. The pattern matching correlation result is shown as a blue solid line in Figure A1(a). In this figure, the high resolution bending angle is shown in grey and shows a sharp transition around the impact parameter $a = 6367$ km. Note that the 1 meter resolution bending angle is used instead of the common low resolution profile which has been filtered with a 200 m window and degraded
$x_b$ precision. The correlation shows a clear peak because the 1-km length step function filters out most of the fluctuations caused by noise, multipath, or highly variable water vapor content close to the ducting layer. The maximum of the correlation function indicated by the dashed line is close to the impact parameter where the sharp transition of the bending angle occurs.

    While this peak can provide the coarse estimation of $x_b$ within 250 m, the length of the step function is very insensitive to the transient behavior of the bending angle. To enhance the precision of the $x_b$ estimate the a second correlation with
30 a shorter step function is used. In this search, a 150 m length step function is used to repeat the correlation with the high resolution bending angle profile. However, when the tangent point lies close to the top of ducting layer, the determination of sharp transition becomes difficult due to fluctuations in the bending angle. On the other hand, the observed bending angle rarely contains large fluctuations below the trapping layer. Therefore, to put more weight in correlation at the lower half of

the step function it has been modified into an asymmetric shape, with a value +1 for lower the 90 meters and a value -1 for the higher 60 meters, which extends the lower part while shrinking the upper part. In addition, the bending angle has been de-trended from the exponential fitting function before applying the second correlation to simplify the profile which can focus on the transition due to ducting instead of normal refractivity increases. The result of the second correlation is shown in Figure

A1(b) as the green line. Due to the shorter integration period the second correlation has higher variability than the first one. The peak of the second correlation is then searched for over the range between -250 m and 250 m relative to the maximum of the first correlation function that covers the impact parameter where the sharp transition could occur. As Figure A1(b) shows, the second correlation peak location, shown in a green line, can clearly determine the location of the sharp transition to be used as the estimated $x_b$. Using this method, $x_b$ can be determined with an uncertainty of less than 50 m.

*Acknowledgements.* This research was supported by an appointment to the NASA Postdoctoral Program at the Jet Propulsion Laboratory, administered by Universities Space Research Association under contract with NASA. Partial support for Feiqin Xie was provided by the NASA grant NNX15AQ17G. We would also like to thank the following: thank Dr. F. J. Turk for helpful discussions; Loknath Adhikari who provided the collocated radiosonde data with AMSR-E and GPS-RO; ERA Interim reanalysis profiles were obtained from the European Center for Medium Range Forecasts (ECMWF). AMSR-E data were obtained from National Snow and Ice Data Center. Radiosonde data

were accessed from NCAR-EOL.

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

| Case Number | RAOB-AMSR | | RAOB-RO | |
|---|---|---|---|---|
| | S | T | S | T |
| 1 | 4.7 km | 0.18 hr | 233.7 km | 1.18 hr |
| 2 | 2.3 km | 1.61 hr | 225.6 km | 0.45 hr |
| 3 | 3.8 km | 1.43 hr | 10.7 km | 1.18 hr |
| 4 | 2.2 km | 0.26 hr | – | – |
| 5 | 431.2 km | 0.34 hr | – | – |
| 6 | 616.1 km | 0.52 hr | – | – |
| 7 | – | – | 262.8 km | 1.22 hr |
| 8 | – | – | 259.7 km | 1.15 hr |
| 9 | – | – | 281.8 km | 0.6 hr |
| 10 | – | – | 292.9 km | 1.28 hr |
| 11 | – | – | 192.8 km | 2.63 hr |

**Table 1.** The spatial and temporal distance between different observation methods for each collocated case analyzed in this paper. The second column are the differences between the RAOB and its closest AMSR-E measurement location in the simulation results, while the third column are the the differences of the actual RO tangent point location and its closest RAOB and measurement. The RAOB observation in cases 1 to 3 are repeated in both simulation and the actual data analysis.

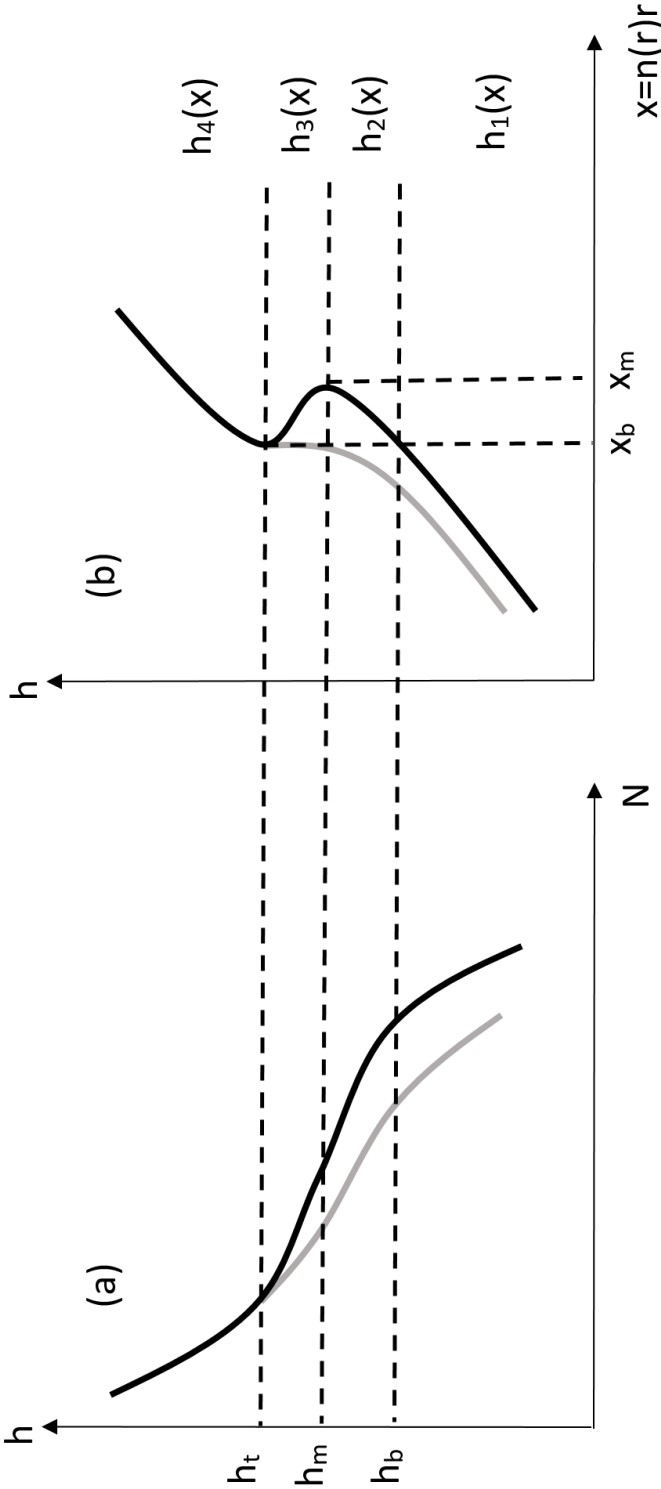

**Figure 1.** The illustration of the corresponding refractivity profile (a) and the $h(x)$ function (b) when ducting occurs. The true profiles are shown in black and the profiles acquired by GNSS-RO Abel retrievals are shown in grey. The large refractivity gradient between $h_m$ and $h_t$ causes the negative slope of $h_3(x)$ and multi-valued function $h(x)$ between $x_b$ and $x_m$

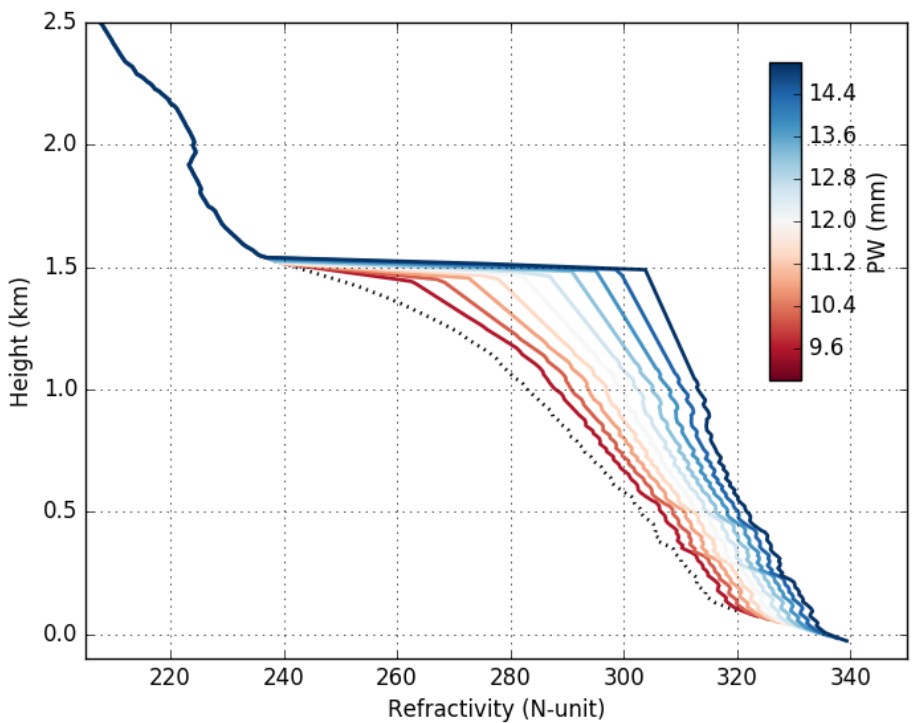

**Figure 2.** The family of refractivity profiles calculated from single simulated GNSS-RO bending angle profile (dotted line) using equation (9) and the parametrization method described in Section 2.2.1. Each profile corresponds to a distinctive PW value, which can be used as a constraint for GNSS-RO retrievals within the boundary layer.

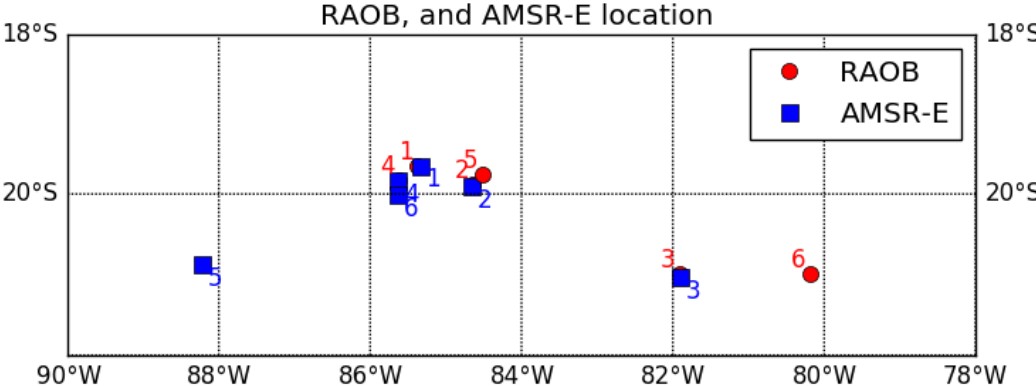

**Figure 3.** The map of the 6 collocated RAOB and AMSR-E measurements in the VOCALS campaign. The spatial and temporal differences for all 6 cases are listed in the Table 1.

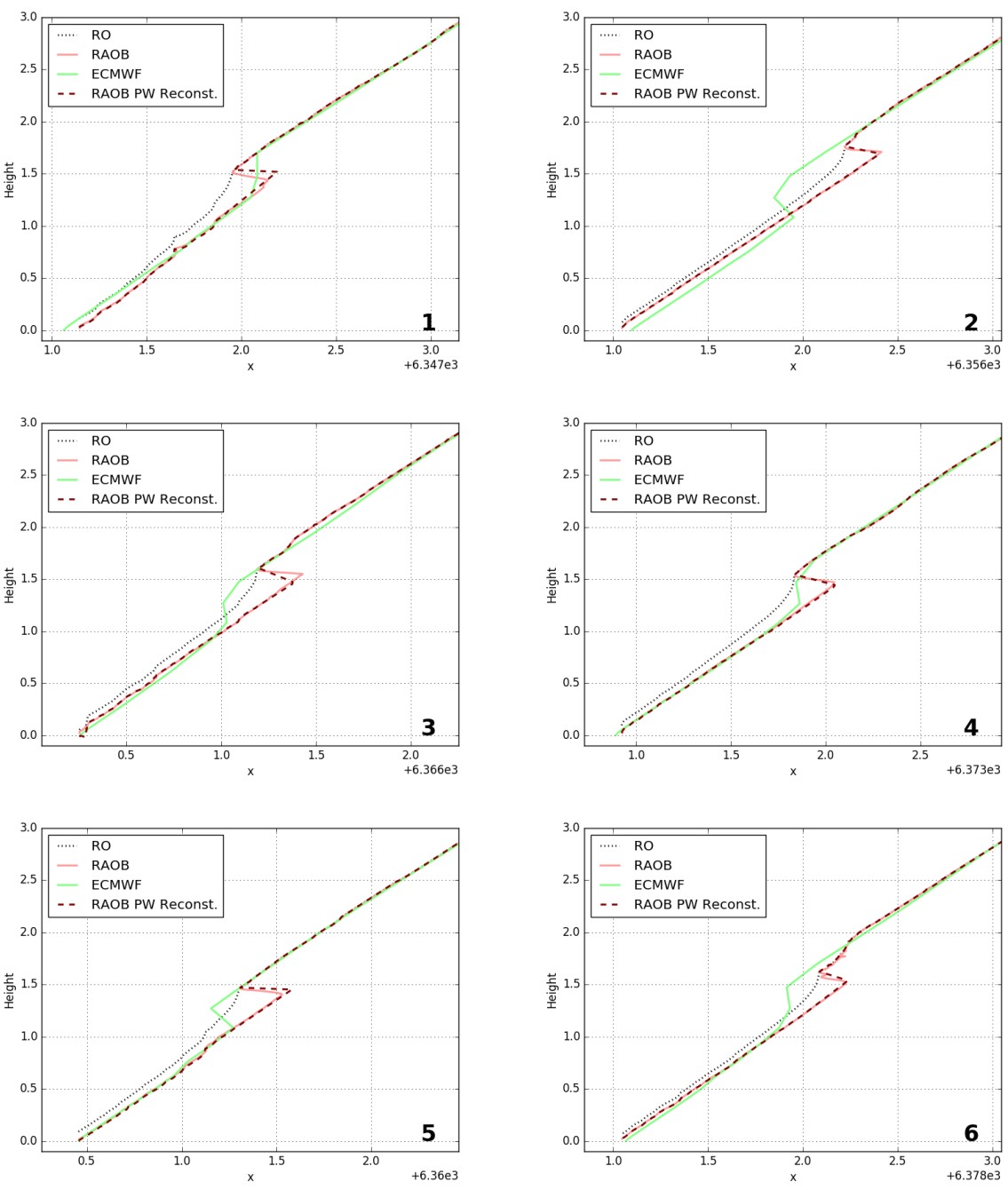

**Figure 4.** The x-h relationship of simulated RO, radiosonde measurements, ECMWF analysis, and reconstruction using RAOB and ECMWF computed PW for the 6 collocated cases. The case numbers are put in the lower-right corner. As shown in the figures the RO simulations will maintain one-to-one relationship between x and height when ducting happens, which cause negative bias compared to the RAOB results. The proposed method can reconstruct the bi-linear shape inside the trapping layer and correct the $N$-bias below.

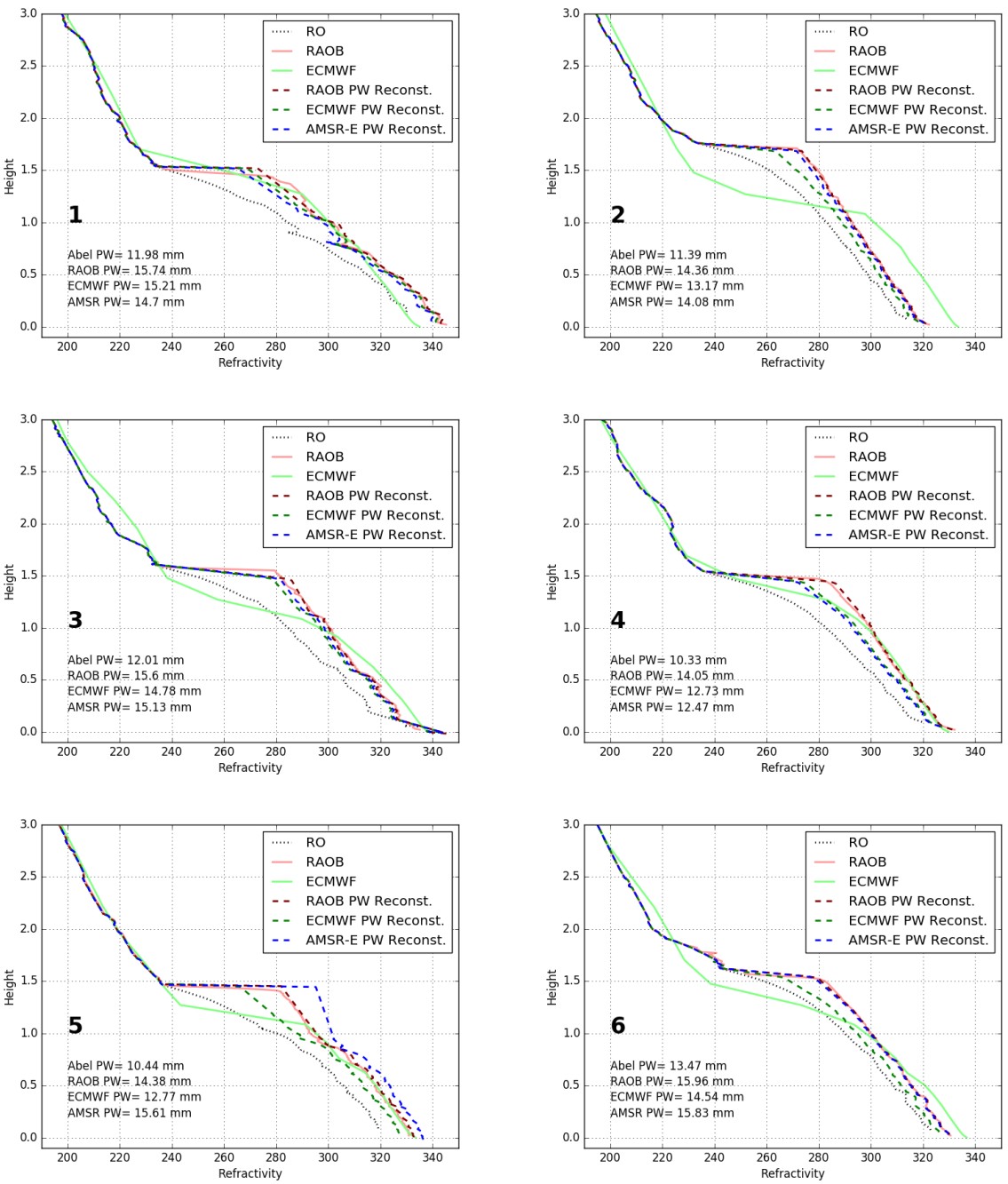

**Figure 5.** Refractivity profiles from simulated RO, radiosonde measurements, ECMWF analysis, and reconstruction using RAOB, ECMWF computed PW and AMSR-E ~~measurement~~ retrievals for the 6 collocated cases. The case number is found on the left side of each panel. It can be observed that the ducting layer height in the ECMWF model is mostly lower than the one measured by radiosondes. The RO optimal estimation results, which correspond to different PW sources, can correct the $N$-bias with a higher amount of water vapor content measured by other techniques.

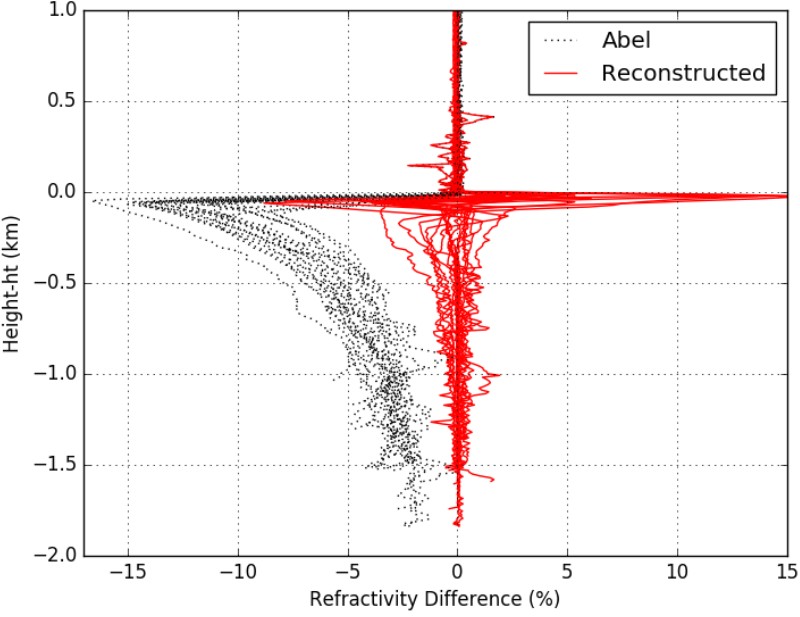

(a) RAOB PW reconst. - RAOB vs Abel - RAOB

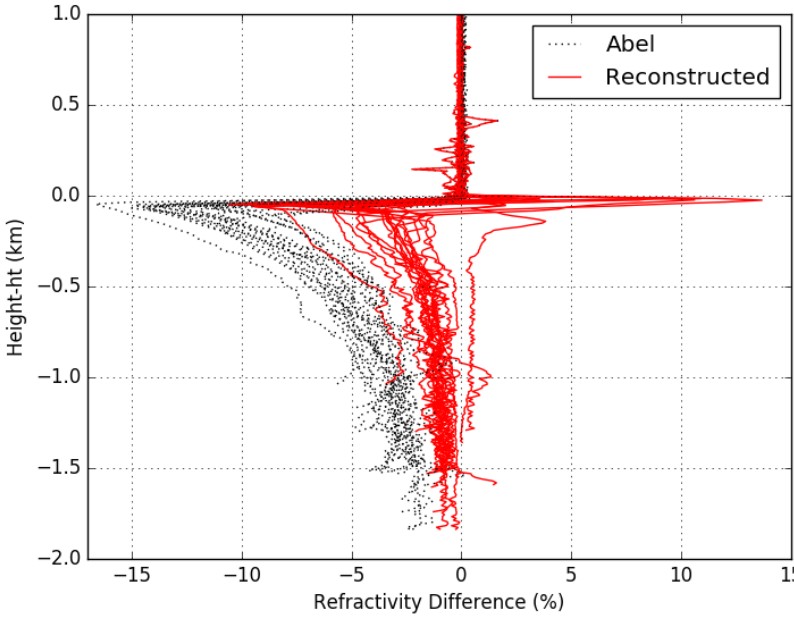

(b) ECMWF PW reconst. - RAOB vs Abel - RAOB

**Figure 6.** The refractivity differences between the RAOB profiles, the Abel-retrieved and the reconstructed profiles for the 19 simulation cases with single ducting layer in the VOCALS campaign using the PW from RAOB (a) and ECMWF (b). While the proposed method can correct the $N$-bias using both PW, the reconstructed profiles show higher variance and are slightly biased below the trapping layer with ECMWF PW.

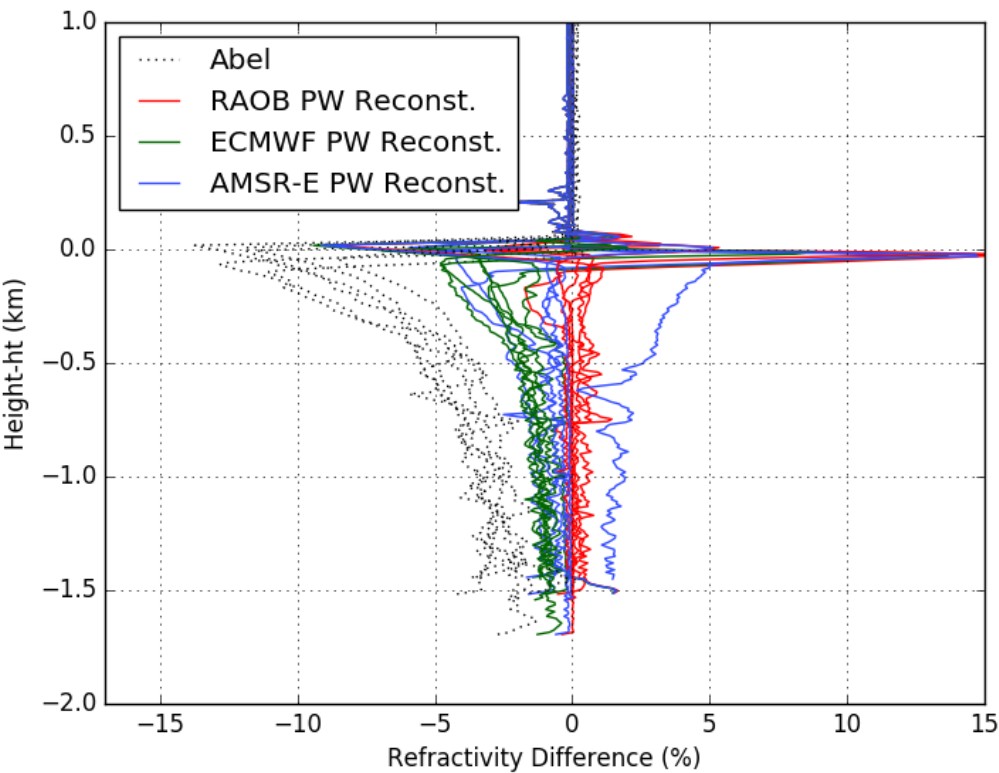

**Figure 7.** Refractivity differences of the Abel-retrieved and the reconstructed profiles from different PW sources compared to the original RAOB profiles in the 6 simulated cases. It can be seen that the results using ECMWF PW are negatively biased (-1% to -5%) while the collocated AMSR-E reconstructed results more closely align with the RAOB profiles. An outlier of the AMSR-E reconstruction shown in the figure is the case 5, which was measured 431 km away from the corresponding RAOB case.

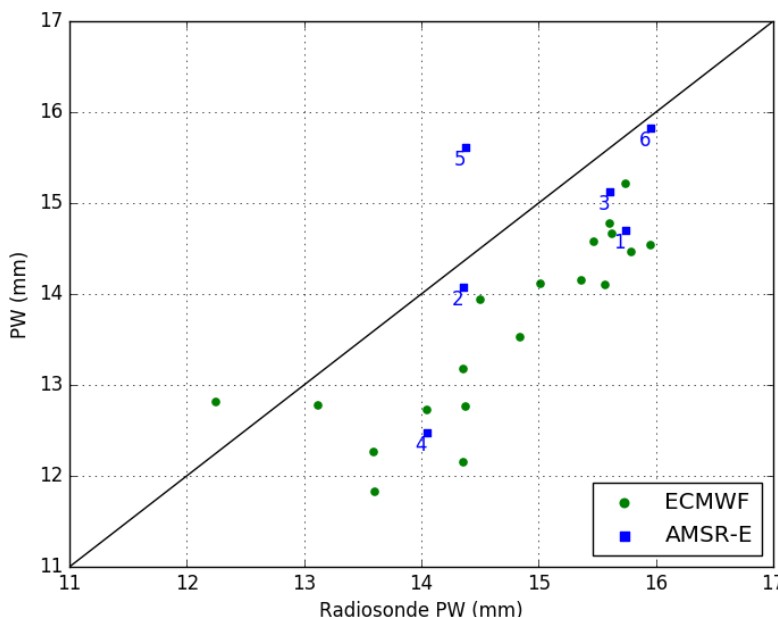

**Figure 8.** PW scatter plot for simulation results from different sources. All 19 RAOB cases PW are compared with the collocated ECMWF and AMSR-E PW in this figure. The x-axis is the RAOB PW, and the black line is the one-to-one line. ECMWF PW are systematically less (1 to 2 mm) than RAOB PW and cause negative biases in reconstructed profiles.

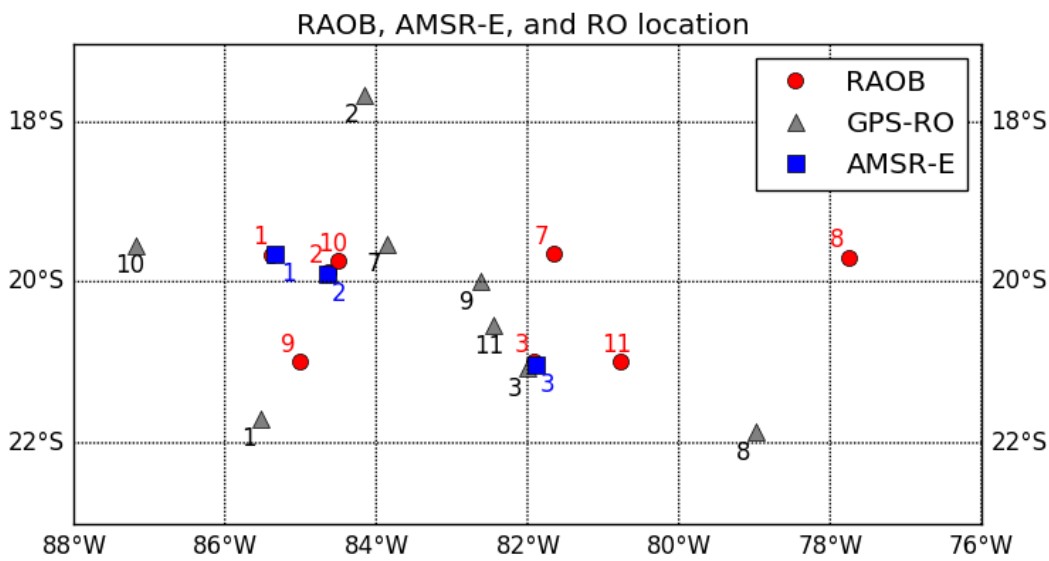

**Figure 9.** The map of the 8 collocated RAOB and COSMIC GPS-RO measurements in VOCALS campaign. Three AMSR-E measurement locations, which coincided with RAOB measurements in the first 3 cases, are shown in blue squares. The distance between the RAOB (and AMSR-E) and the corresponding RO cases are 250 km in average. The temporal difference for all 8 cases is within 3 hours.

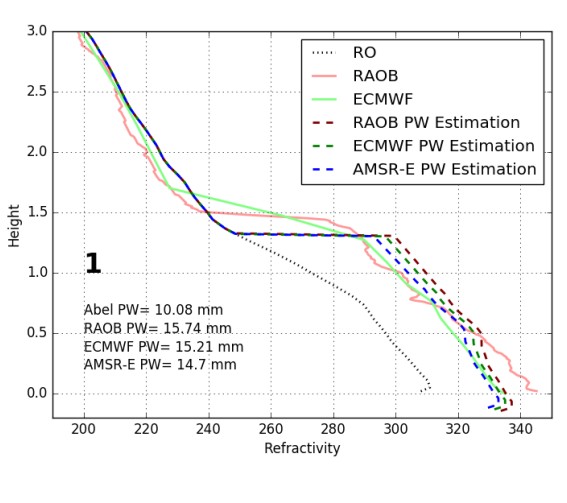

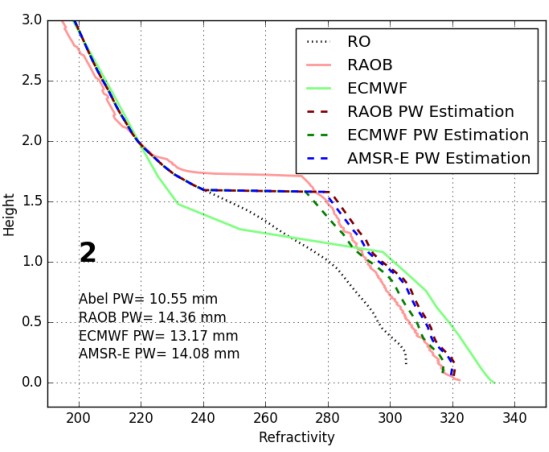

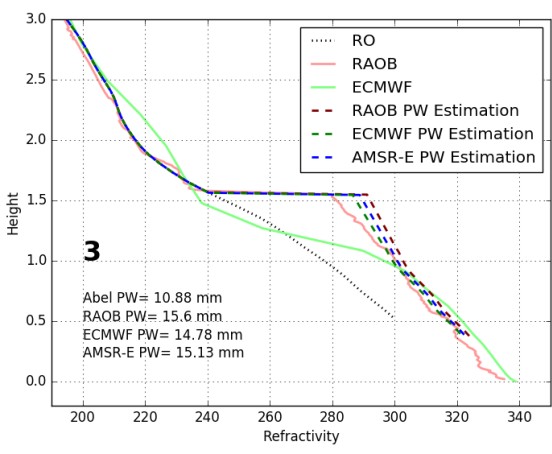

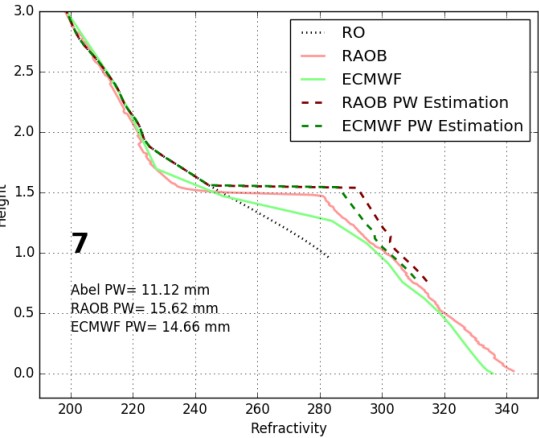

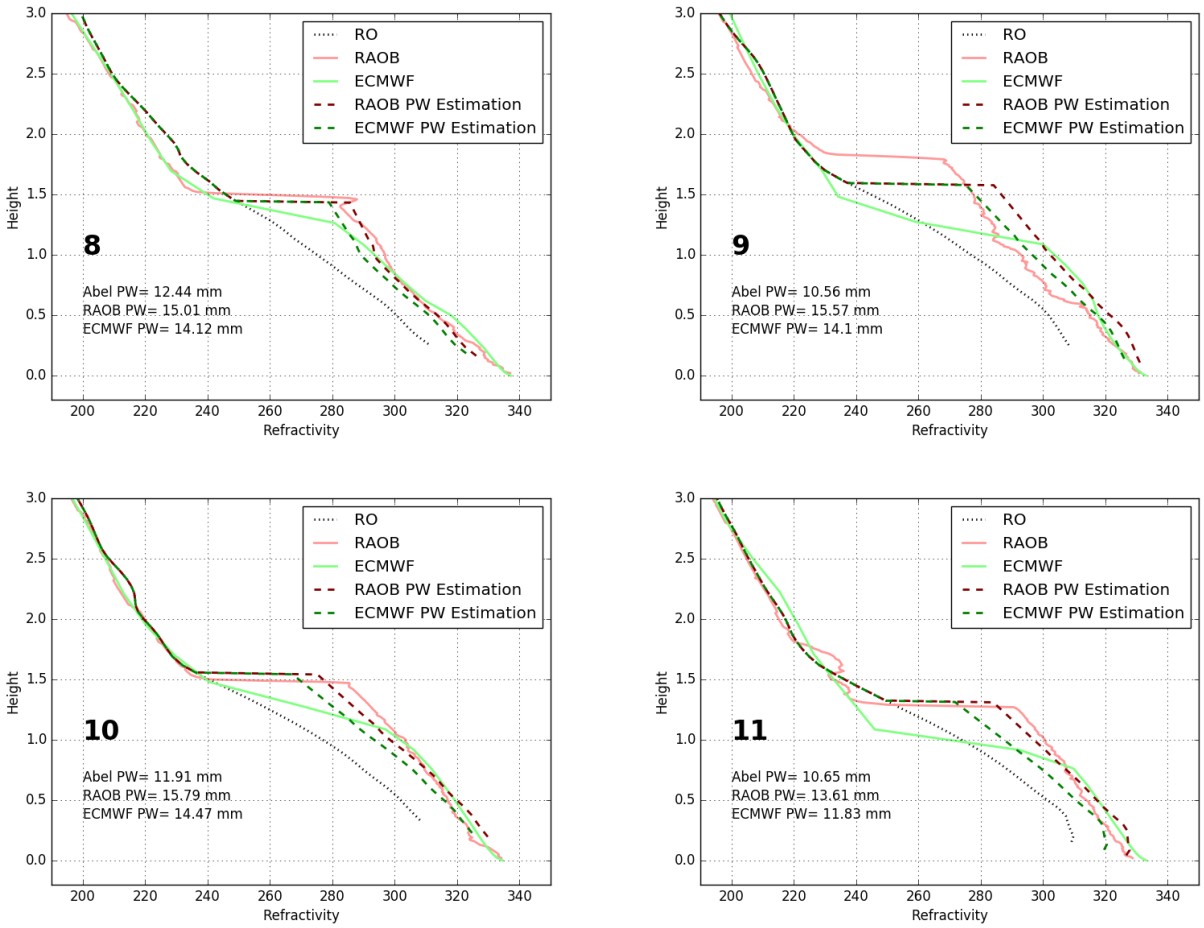

**Figure 10.** Refractivity profiles from the actual RO (dotted lines), collocated radiosonde measurements (solid red lines), ECMWF analysis (solid green lines), and reconstruction using RAOB (red dashed lines), and ECMWF computed PW (green dashed lines) for the 8 collocated cases in VOCALS campaign. The reconstructed profiles using collocated AMSR-E measurements are also shown (blue dashed lines) in the first 3 cases. All case numbers are in the left side of each panel. Although the ducting layer heights in most cases are different from the RAOB profiles due to the distance and different footprints between the two observations, the RO reconstruction results are still able to correct the $N$-bias below the trapping layer with higher amount of water vapor content measured by the AMSR-E or ECMWF model.

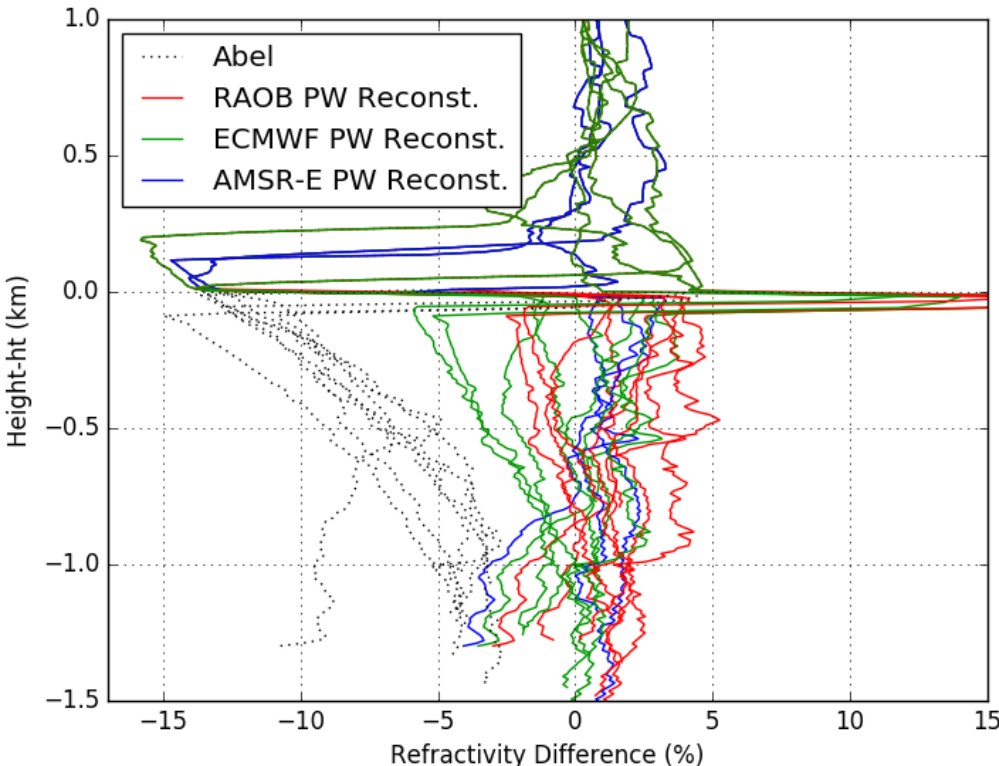

**Figure 11.** The refractivity differences of the Abel-retrieved and the reconstructed profiles from different PW sources compared with the original RAOB profiles in the 8 actual data cases. Compared to the 15% negative bias below the height $h_t$ from the Abel inversion results, the reconstructed profiles utilizing closed PW sources can limit the error within 5%. The large negative bias and variance above $h_t$ are mostly due to the spatial and temporal distances between the RO and RAOB, which are normally more than 200 km and 1 hour apart. As the results shown in the simulation, negative biases can still be observed in the ECMWF PW reconstructed profiles in actual data cases.

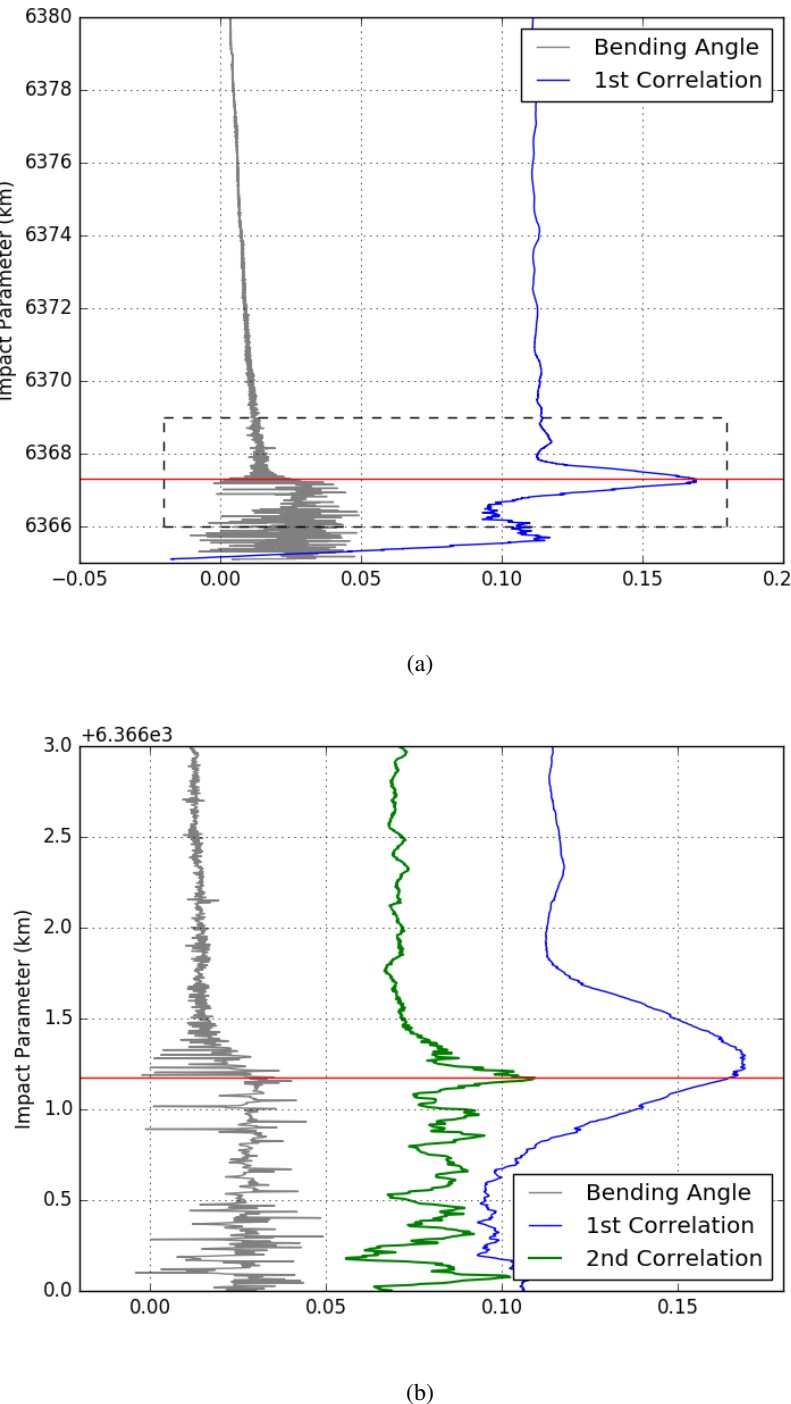

(a)

(b)

**Figure A1.** Ducting layer height determination using bending angle profiles. The correlation result with the long step function is shown in (a) blue line, which has been scaled and shifted for demonstration. The peak location identifies the approximated ducting layer height. The correlation result with the short step function is shown in (b) with a green line. Panel (b) is the zoom-in image of the dashed-line box shown in panel (a). The corresponding impact parameter of the sharp bending angle transition can be found at the location of the second correlation peak within the range of the first correlation hump (±250 m). The correlation results in both panels are scaled down and shifted to fit into the figures.