# Peer review of "Correcting negatively-biased refractivity below ducts in GNSS radio occultation: An optimal estimation approach towards improving planetary boundary layer (PBL) characterization"

_Atmospheric Measurement Techniques, 2017_

## Referee Comment (RC1) · Anonymous Referee #1 · 24 May 2017

General Comments

This paper sets out a new retrieval method for GPS-RO refractivity below a ducting layer. This problem is known to be ill-posed, and therefore a-priori information is required to provide a solution. The new approach uses ECMWF analyses and AMSR-E preciptable water retrievals (not measurements!) to make the problem well-posed. The authors present simulated retrievals errors and apply the new approach to COSMIC profiles. As might be expected, the errors with real COSMIC data are larger when verified against radiosonde observations. The paper requires major revision before

publication.

There is no discussion on the impact of horizontal refractivity gradients errors on the retrieval performance. It would be useful for the authors to consider the recent Zeng et al (2016, Appendix A) paper (http://www.atmos-meas-tech.net/9/335/2016/) that show that horizontal gradients in the ionosphere can lead to features being assigned the wrong height. More generally, if atmospheric and ionospheric horizontal gradients are causing an impact parameter error, da, the resulting radius or height error, dr, is

$dr = da/(n+r.(dn/dr))$

where n is the refractive index, and r is the radius. The key point here is that impact parameter errors are amplified when mapped to radius, and this is particularly problematic for ducting conditions where $r.(dn/dr) \sim 1$. How does this affect your interpretation?

Secondly, in the context of NWP assimilation, if an NWP system is assimilating refractivity/bending angles down close to the ducting later, and is also assimilating other radiances like AMSR-E and, would the retrieved refractivity profiles below the ducting layer provide any extra information? If the authors argue that the retrieved refractivity is not intended for NWP assimilation, that is reasonable but it should be stated in the text.

Specific comments

Page 4, Line 28, "Able" should be "Abel".

Page 8, The AMSR-E PW values are not "measurements". They are retrieved quantities that will depend on a-priori information. Please correct this throughout the paper. What a-priori is used in the AMSR-E retrievals? EG, do they have to assume a temperature profile?

Page 8, ECMWF analysis information. Are you using the 137 vertical levels, horizontal reolution, etc? Please give details.

Page 8, equation 9. The temperatures in this equation should be virtual temperatures? Typo or bug in the retrieval? More generally, are you using virtual temperatures when you compute the height of the ECMWF levels?

Page 9, $C_y$ in equation should include a forward model error term. EG, caused by assuming ECMWF tmperatures are "true" in eq.8, assuming the $q(z)$ is constant etc. Have you estimated it?

Section 3.

When generating the observed bending angle from the raob, I assume you integrate eq.2 or 3? Please state this, and give more details. It should be emphasised that horizontal gradient errors are neglected in simulations in this section. Have the raobs been assimilated at ECMWF - ie, the raobs and analysis could be correlated? It might be interesting to see if the ECMWF forecasts at the raob locations look very different.

Section 4

Page 12. Line 19. "no double or complex structure inside the trapping layer". Please explain what is being screened out here, and how often it happens.

Page 12, last line: I suggest that the 200 m difference between estimated x_b and the corresponding radiosonde information could be caused by the variation of n.r.sin(phi)(=a) along the ray path. This a well known consequence of horizontal gradients. Have you investigated this by looking at gradients in the analysis fields along the ray paths?

Figure 12. Suggest rename it Fig. A1, because its only referenced in the appendix.
* * *

---

## Referee Comment (RC2) · Anonymous Referee #2 · 28 Jun 2017

The manuscript addresses the negative bias that is often found during the retrieval of refractivity, which is known to be related to the occasional presence of superrefractive layers in the low troposphere. The manuscript proposes a procedure to identify the presence of these layers, and to estimate the refractivity within the layer, using external information. This should allow an improved retrieval of the refractivity profile from a bending angle profile.

The subject is interesting, and this exercise may be useful. However, my main concern is that the authors should better explain why it is important to, for instance, try to address this negative bias, rather than to accept that the retrieval of refractivity from bending angle presents limitations and is an ill-posed problem below superrefractive layers. As with other underdetermined problems, adding sufficient information eventually provides a closure. The information that best provides this closure is most likely user-dependent. The authors specifically propose the integrated precipitable water (PW) as closure. Although PW may be often available, it is a source external to GNSSRO. The authors should explain why PW is to be preferred over other quantities that may also be equally available, and are also external, or why the retrieval of the refractivity profile is still important, when it can only be carried out requesting external information.

Specific comments:

P1L16: "that couples", to for instance ", and couples"

P1L23: "technique precisely" to "technique that precisely"

P2L14: "transceiver geometry". Current GPSRO does not use transceivers. The transmitter never receives, and the receiver never transmits.

P2L16: "information inside the ducting layer will be missing". It is not really missing. GNSSRO does not provide it.

P2L19: "To mitigate the N-bias". It should be explained why it must be mitigated, rather than accepted.

P2L30: "Measurements of PW". There are very few actual measurements of PW available. Information of PW does exist, but most are retrievals, estimations, or background model information. Given that PW is often not the actual source, why should PW be used, rather than the original measurements?

P3L3: "For a single ray path...": This is ok only in a spherically symmetric refractivity field.

P3L5: "n is the refractivity index". It is the refraction index.

P3L6: "assumption of a spherically symmetric": The assumption is already necessary above.

P3L17: "measured bending angle". Should be "measured bending angle profile".

P4L18: "signal with a tangent point inside the trapping layer cannot be received". I would not say that. There is no signal whose TP is inside the trapping layer, not an existing signal that cannot be received. Tangent points are either above or below.

P4L21: "multiple values". This is not a numerical problem, and there is nothing unphysical. Eq (2) represents a function that is not defined for all $r\_t$. This is related to, but not caused by, the existence of multiple heights with the same x.

P4L22: "retrieved". Bending angle is not being retrieved here. It is being evaluated.

P4L23: "bending angle of these rays can still be retrieved through the regular Abel inversion". The bending angle can be evaluated if we know the refractivity profile. But the Abel inversion (Eq 4) retrieves the refractivity. And this cannot be retrieved once below the superrefractive layer.

P5L3: "Information loss". It is not lost. It is not being gained. Also later in P13L18.

P5L4: "the most significant negative bias". Please clarify the sentence.

P5L23: "error parameter" should be "error in the parameter".

P7L10: "sensitive to the mis-modeling of". If this is true, then the procedure is weak. The meaning of the rest of the paragraph is unclear. Please rewrite.

P8L4: "PW meaurements". I understand that the measurements are brightness temperatures related to PW, not PW itself. Also, ground-based GNSS provides ZTD, with PW being derived only through the approximate subtraction of the hydrostatic delay, which must itself be estimated from further measurements or external information.

P13L19: Profiles of retrieved bending and retrieved refractivity still have a 1-1 correspondence. The lost 1-1 is between the atmospheric refractivity and the retrieved bending.

P13L28: "PW measurements". The measured quantity is a brightness temperature. But given that external information must be used, why PW? Why from AMSR-E? Why not from a background model? Why not some other quantity or quantities from a background model?

---

## Author Comment (AC2) · 10 Aug 2017

**Reviewer #2**

The subject is interesting, and this exercise may be useful. However, my main concern is that the authors should better explain why it is important to, for instance, try to address this negative bias, rather than to accept that the retrieval of refractivity from bending angle presents limitations and is an ill-posed problem below superrefractive layers. As with other underdetermined problems, adding sufficient information eventually provides a closure. The information that best provides this closure is most likely user dependent. The authors specifically propose the integrated precipitable water (PW) as closure. Although PW may be often available, it is a source external to GNSSRO. The authors should explain why PW is to be preferred over other quantities that may also be equally available, and are also external, or why the retrieval of the refractivity profile is still important, when it can only be carried out requesting external information.

<Response>

Thank you for the valuable comment. These issues are addressed in the following responses.

Specific comments:
P1L16: "that couples", to for instance ", and couples"

<Response>

Thank you for the suggestion, the change is made.

P1L23: "technique precisely" to "technique that precisely"

<Response>

Thank you for the suggestion, the change is made.

P2L14: "transceiver geometry". Current GPSRO does not use transceivers. The transmitter never receives, and the receiver never transmits.

<Response>

Thank you for the suggestion, this sentence has been changed to:

Due to the **transmitter and receiver** geometry of GNSS-RO, the tangent point…

P2L16: "information inside the ducting layer will be missing". It is not really missing. GNSSRO does not provide it.

<Response>

We agree that the information inside the ducting layer is not "missing". We meant the ducting information does not exist in bending angle measurement. To avoid the confusion, we modify this sentence to the following at P2 L14:

**As a result, the GNSS-RO bending angle measurements will loss the information inside the ducting layer, in which cannot be recovered using solely GNSS-RO observations.**

P2L19: "To mitigate the N-bias". It should be explained why it must be mitigated, rather than accepted.

<Response>

In PBL research, characterizing the vertical structure of the PBL is important because it can be related to different atmospheric processes associated with the PBL [Ao et al., 2012]. However, although the weather analysis incorporated both GNSS-RO and AMSR-E measurements, the systematic low bias of the ECMWF PBL height [Xie et al., 2012] shows that the observations were not optimally assimilated into the model below the ducting layer. Therefore, we argue that developing an independent bias-mitigation retrieval process outside of NWP data assimilation is valuable.

To clarify this idea, we added the following sentences right before the paragraph:

**Correcting the N-bias within the PBL is essential towards the use of RO in studying the vertical structure within the PBL. While the weather analyses can assimilate RO bending angles, which are unaffected by the refractivity bias caused by ducting, it is not clear that the analyses can optimally handle these high vertical resolution measurements. In addition, the analyses may be strongly affected by bias in the model, as evidenced by the low PBL height over the stratocumulus regions (Xie et al., 2012). Therefore, it is of great scientific interests to retrieve an unbiased PBL refractivity based on observations only.**

P2L30: "Measurements of PW". There are very few actual measurements of PW available. Information of PW does exist, but most are retrievals, estimations, or background model information. Given that PW is often not the actual source, why should PW be used, rather than the original measurements?

P8L4: "PW meaurements". I understand that the measurements are brightness temperatures related to PW, not PW itself. Also, ground-based GNSS provides ZTD, with PW being derived only through the approximate subtraction of the hydrostatic delay, which must itself be estimated from further measurements or external information.

P13L28: "PW measurements". The measured quantity is a brightness temperature. But given that external information must be used, why PW? Why from AMSR-E? Why not from a background model? Why not some other quantity or quantities from a background model?

<Response>

Thank you for this valuable feedback. First, as the other reviewer also pointed out, the PW used in this article is not a "raw measurement" but a "retrieval result". We agree with your comment and made the changes throughout the article.

Second, there are several reasons to choose PW over other physical quantities:

    i)        PW is widely available through a variety of measurement techniques.
    ii)      Most of the PW is concentrated within the PBL.
    iii)     PW value of a GNSS-RO refractivity profile can be calculated easily.

Indeed, some alternative measurements or retrievals are also appropriate in our optimization scheme. For example, the 3-D water vapor from AIRS and cloud top temperature from MODIS could also be potential candidates to distinguish different refractivity profiles. But in this study, PW is used and proved to be a feasible solution for our purpose.

To emphasize the advantage of using PW, we added the sentences at each paragraphs of P8:

**First, most of the water vapor in the atmosphere is located within the PBL so that accurate PW observations can provide extra information below the ducting layer to assist GNSS-RO retrievals.**

**Second, PW are available globally from a variety of sensors (Millan et al, 2016).**

**Third, PW value of a candidate refractivity profile can be easily calculated to compare with the PW observations.**

Third, although using the quantity estimated by the background models is convenient, we prefer a measurement-based approach when possible. As shown in our results (Figure 5 and Figure 10), the structure of PBL estimated by ECMWF analysis were very different from the ones observed by RAOB. In fact, this systematic low bias of the ECMWF PBL height has been reported (e.g. [Xie et al., 2012]). Therefore, the PW and other quantities provided by the background model may not be accurate enough, and the use of independent sensing sources might be preferred in some cases. The detailed comparison between different PW sources can be found in the discussion between L21 and L26 in P12.

To better illustrate this reason, we added the following sentences at P11 L17:

**The statistically low PBL heights in ECMWF, which were extensively observed in the region, implies an erroneous refractivity profile below the ducting layer. This difference has been attributed to the model physics and assimilation process limitations (Xie et al., 2012). Even though ECMWF and other NWP system assimilate both GNSS-RO bending angles and AMSR-E radiances, it is not clear that the full vertical resolution of the measurements can be taken into account. Thus an independent, unbiased, refractivity retrieval outside of NWP data assimilation systems remains extremely valuable.**

P3L3: "For a single ray path: : :": This is ok only in a spherically symmetric refractivity field.

<Response>

Thank you for your comment. We modified this sentence to:

**Under the assumption of a spherically symmetric atmosphere, the impact parameter 'a' of a single ray path can be defined as:**

P3L5: "n is the refractivity index". It is the refraction index.

<Response>

Thank you for the suggestion; the change has been made.

P3L6: "assumption of a spherically symmetric": The assumption is already necessary above.

<Response>

Thank you for your comment. We removed this assumption in this line:

 the accumulated bending angle of a GNSS-RO ray path can be calculated…

P3L17: "measured bending angle". Should be "measured bending angle profile".

<Response>

Thank you for the suggestion; the change has been made.

P4L18: "signal with a tangent point inside the trapping layer cannot be received". I would not say that. There is no signal whose TP is inside the trapping layer, not an existing signal that cannot be received. Tangent points are either above or below.

<Response>

We agree with this comment. Therefore, we modified the following sentence as:

Because of the geometry of GNSS RO, in which both the transmitter and the receiver are located outside the Earth atmosphere, **the tangent points of the received signals will not appear inside the trapping layer.**

P4L21: "multiple values". This is not a numerical problem, and there is nothing unphysical. Eq (2) represents a function that is not defined for all r_t. This is related to, but not caused by, the existence of multiple heights with the same x.

<Response>

In this paragraph we planned to use Eq (2) to illustrate the idea of "locating the tangent points inside the trapping layer is unphysical". If we try to calculate the non-existing bending angle inside the trapping layer, which is the range of not defined as pointed out, the integrand value of Eq (2) becomes invalid (complex). This invalidity is mathematically caused by the existence of multiple heights with the same x.

To clarify, we modify this paragraph as follows:

**This gap can be noticed by examining equation (2). When evaluating the bending angle with (2) inside the trapping layer, $h_b < r_t - r_e < h_t$, the term $n(r)r$ above the height $r_t$ becomes less than $n(r_t)r_t$ because of the negative gradient of $x$ between $h_m$ and $h_t$. It would lead to a negative value inside the square root in equation (2) and the solution is a complex number for the bending angle inside the trapping layer which is unphysical.**

P4L22: "retrieved". Bending angle is not being retrieved here. It is being evaluated.

\<Response\>

Thank you for the suggestion; we changed the word "retrieved" to "evaluated".

P4L23: "bending angle of these rays can still be retrieved through the regular Abel inversion". The bending angle can be evaluated if we know the refractivity profile. But the Abel inversion (Eq 4) retrieves the refractivity. And this cannot be retrieved once below the superrefractive layer.

\<Response\>

Thank you for pointing this out; we accidentally combined two sentences in a wrong way. This sentence has been updated to:

**bending angle of these rays can still be calculated by GNSS-RO**

P5L3: "Information loss". It is not lost. It is not being gained. Also later in P13L18.

\<Response\>

We deleted this phrase in P5L3 and changed the "information loss inside the trapping layer" to "**the lack of bending angle information in the trapping layer** " in P13L18.

P5L4: "the most significant negative bias". Please clarify the sentence.

\<Response\>

We rewrote this sentence as follows at P5 L8:

**Among all the refractivity solutions corresponding to the same bending angle profile, the one retrieved by the standard Abel introduces the largest negative bias.**

<Response>

P5L23: "error parameter" should be "error in the parameter".

Thank you for the suggestion, the change is made.

<Response>

P7L10: "sensitive to the mis-modeling of". If this is true, then the procedure is weak. The meaning of the rest of the paragraph is unclear. Please rewrite.

In this paragraph we meant to explain the "top" of h1(x) calculated by equation (9), rather than the whole profile, is sensitive to the mis-modeling. To clarify the idea, we rewrite the whole paragraph as follows at P7 L19:

**First, the highest 100 m of h1(x) will be replaced by the linear extrapolation from below, its slope is determined by linear regression between 100 m and 200 m below the height hb. The reason is that the top of the analytical solution calculated by equation (9) is sensitive to the mis-modeling of h2(x) and h3(x), which are assumed to be bi-linear segments between hb and ht. Spurious spikes and fluctuations are found at the top 100 m of the function h1(x) if the straight line assumption inside the critical layer is violated by the data, or the parameters, e.g. hb, are not appropriately estimated from Equation (12). While these fluctuations in h1(x) are usually small (<20 m) and typically occurred within 100 m below hb, they can significantly change the slope at hb, which makes the h2(x) value obtained from assumption (A3) inconsistent with having one refractivity for each impact parameter below hb. Therefore, we replaced the top portion of h1(x) with a linear extension of the curve below to remove the fluctuation. Note that the variable hb changes when the top of h1(x) is replaced. To avoid the inconsistency between the new hb and the originally specified hb, xm is chosen over hb as the "free" variable to construct a profile inside and below the critical layer**

<Response>

P13L19: Profiles of retrieved bending and retrieved refractivity still have a 1-1 correspondence. The lost 1-1 is between the atmospheric refractivity and the retrieved bending.

Thank you for the suggestion; we updated the following sentence:
**As the retrieved bending angle loses its one-to-one relationship with the atmospheric refractivity**, the standard Abel inversion will give the refractivity solution with the largest negative bias.

---

## Author Comment (AC1)

**Reviewer #1**

There is no discussion on the impact of horizontal refractivity gradients errors on the retrieval performance. It would be useful for the authors to consider the recent Zeng et al (2016, Appendix A) paper (http://www.atmos-meas-tech.net/9/335/2016/) that show that horizontal gradients in the ionosphere can lead to features being assigned the wrong height. More generally, if atmospheric and ionospheric horizontal gradients are causing an impact parameter error, da, the resulting radius or height error, dr, is dr = da/(n+r.(dn/dr)) where n is the refractive index, and r is the radius. The key point here is that impact parameter errors are amplified when mapped to radius, and this is particularly problematic for ducting conditions where r.(dn/dr)_1. How does this affect your interpretation?

Page 12, last line: I suggest that the 200 m difference between estimated x_b and the corresponding radiosonde information could be caused by the variation of n.r.sin(phi)(=a) along the ray path. This a well known consequence of horizontal gradients. Have you investigated this by looking at gradients in the analysis fields along the ray paths?

<Response>

Thank you very much for this valuable comment. Indeed, the horizontal refractivity gradient is an issue which could cause erroneous ducting layer height estimation. We are now including a note that mentions these sources of error.

Ducting can affect the RO retrieval process in two independent ways: bending angle error due to horizontal refractivity gradient and the ill-posed problem in refractivity determination. In this article, we focus on solving the effects of ducting in retrieving refractivity profiles, rather than addressing the other issues in the bending angle calculation. To the best of our knowledge, all currently used bending angle and refractivity retrieval processes do not address the violation of spherically symmetric refractivity distribution in the ionosphere, and the retrieved results will contain certain degree of error caused by horizontal refractivity gradients. This error should have the same order of impact on the refractivity profiles retrieved by classical Abel-inversion and the proposed reconstruction method.

On the other hand, we strongly agree that horizontal refractivity gradient is an important factor and should be stated in the main text. Therefore, we added the potential effect in P13 L28:

**Another possible cause of x_b discrepancy is the error in GNSS-RO measurement due to horizontal inhomogeneity in the atmosphere and the ionosphere (Zeng et al., 2016). In ducting conditions, this error can be amplified and shift the impact parameter of boundary layer top for more than 100 m. While addressing the horizontal inhomogeneity is beyond the scope of this article, the impact of horizontal refractivity gradient on the reconstruction method can be further investigated in future work.**

Secondly, in the context of NWP assimilation, if an NWP system is assimilating refractivity/bending angles down close to the ducting later, and is also assimilating other radiances like AMSR-E and, would the retrieved refractivity profiles below the ducting layer provide any extra information? If the authors argue that the retrieved refractivity is not intended for NWP assimilation, that is reasonable but it should be stated in the text.

<Response>

While NWP assimilation can incorporate both measurements, the results cannot accurately model the PBL. One example is the systematic low bias of the ECMWF PBL height [Xie et al., 2012], which can also be observed in many cases when compared to the RAOB results as shown in Figure 5 and Figure 10. It appears that the observations of GPS-RO and AMSR-E were not optimally assimilated into the model below the ducting layer, and this could have impacts on cloud evolution simulations. Therefore, we argue that it is valuable to develop an independent refractivity retrieval process outside of NWP data assimilation.

To better explain this reasoning, we added the following sentences at P11 L17:

**The statistically low PBL heights in ECMWF, which were extensively observed in the region, implies an erroneous refractivity profile below the ducting layer. This difference has been attributed to the model physics and assimilation process limitations (Xie et al., 2012). Even though ECMWF and other NWP system assimilate both GNSS-RO bending angles and AMSR-E radiances, it is not clear that the full vertical resolution of the measurements can be taken into account. Thus an independent, unbiased, refractivity retrieval outside of NWP data assimilation systems remains extremely valuable.**

Specific comments

Page 4, Line 28, "Able" should be "Abel".

<Response>
Thank you for your comment, the change has been made.

Page 8, The AMSR-E PW values are not "measurements". They are retrieved quantities that will depend on a-priori information. Please correct this throughout the paper. What a-priori is used in the AMSR-E retrievals? EG, do they have to assume a temperature profile?

<Response>

Thank you for the comment, we corrected them in the article. The details of the PW retrieval algorithm from AMSR-E can be reviewed in the following technical report:

***Yoshiaki Takeuchi (2002), Algorithm theoretical basis document (ATBD) of the algorithm to derive total water vapor content from ADEOS-II/AMSR, EORC Bulletin/Technical Report -- Special Issue on AMSR Retrieval Algorithms***

This report has also been added in the reference list of the article. According to this report, no temperature profile is needed for the retrieval. However, the temperature at 850hpa and sea surface level from global analysis is required.

Page 8, ECMWF analysis information. Are you using the 137 vertical levels, horizontal resolution, etc? Please give details.

<Response>

Thank you for your suggestion, we added the ECMWF information at P8 L24:

**The high resolution ECMWF analysis data (TL799L91) used in this research have 91 vertical levels from the surface to 0.01 hPa and 0.25° horizontal resolution. The data is modeled at every 6 hours and unevenly sampled in vertical space which has higher resolution near surface (~40 m).**

Page 8, equation 9. The temperatures in this equation should be virtual temperatures? Typo or bug in the retrieval? More generally, are you using virtual temperatures when you compute the height of the ECMWF levels?

<Response>

There was no equation (9) on page 8.  We assume that you are referring to equation (15). In that case, T is the temperature instead of the virtual temperature [Kursinski and Hajj, 2001]. In this equation $\bar{m}$ is the mean molecular mass taking both dry air and vapor into account:

$$\bar{m} = m_d \frac{p - e}{p} + m_v \frac{e}{p}$$

And in the direct method we have to calculate $\bar{m}$ at each step of iteration using this equation along with the evolving p and e information. To clarify this we added this equation to the manuscript and more description of the direct method:

$\overline{m}$ is the mean molecular mass of atmosphere which takes both dry air and vapor into account:

$$\overline{m} = m_d \frac{p-e}{p} + m_v \frac{e}{p} \tag{16}$$

where $m_v$ and $m_d$ are the molecular mass of dry air (~28.97 g/mol) and water vapor (~18.02 g/mol), respectively. Using the equation (15) along with the refractivity equation (5) one can solve the water vapor pressure profile e iteratively by updating $\overline{m}$ at each step and the convergence at each height interval can be reached in one or two iterations.

In addition, the ECMWF height is also not computed with virtual temperatures.

Page 9, C_y in equation should include a forward model error term. EG, caused by assuming ECMWF temperatures are "true" in eq.8, assuming the q(z) is constant etc. Have you estimated it?

<Response>

We plan to perform a more detailed error analysis in a follow-on study. However, a simple test is provided to show the rough estimate of the variation in calculated PW results.

A case of RAOB specific humidity (q) profile is used for simulation (Figure R1). In [Kursinski and Hajj 2001], the 1-σ error of the retrieved q using direct method is estimated as 0.2 g/kg. While these errors cannot be easily modeled, we simulate the sum of the forward model error as the non-biased random noise of 0.5 g/kg. The profile with the noise added is shown in Figure R2. We generated 50 noisy q profiles and calculate PW for each of them. The resulting PW standard deviation of these 50 cases is ~0.11mm. Since we conservatively set our PW σ margin as 1 mm in C_y (including the AMSR-E retrieval σ = 0.6mm), the forward modeling error should already been well-considered.

The reason that PW can contain such a small error is because it is calculated by integration, which can be viewed as a low-pass filter to block complex humidity features and uncertainties. However, since simulating the error as a Gaussian noise may not be accurate enough in practice, a more detailed error analysis has to be further investigated in the future. In this article, we also added the forward model error in the sentence of C-y calculation:

**The AMSR-E PW retrieval contains an error of ~0.6 mm, but additional errors could rise from RO - AMSR-E collocation distances and forward modeling. Therefore, the conservative PW margin of 1 mm is used as the uncertainty of the PW observation in the C_y matrix.**

[Figure]

Figure R1.  The specific humidity profile from one of VOCALS RAOB cases

[Figure]

Figure R2.  The noisy ($\sigma = 0.5$) specific humidity profile from the same case of Figure R1

Section 3.
When generating the observed bending angle from the raob, I assume you integrate eq.2 or 3?
Please state this, and give more details. It should be emphasized that horizontal gradient errors
are neglected in simulations in this section.

<Response>

Thank you for the suggestion, we added more details in the following sentences at P11 L8:

**While x is not monotonically increasing in the RAOB refractivity profiles, the forward
calculation of equation (2) should be used in here to generate the RO bending angle. Note
that the potential errors caused by horizontal refractivity gradient are neglected in the
bending angle simulation.**

Have the raobs been assimilated at ECMWF - ie, the raobs and analysis could be correlated? It
might be interesting to see if the ECMWF forecasts at the raob locations look very different.

<Response>

No, the radiosonde data from VOCALS campaign were not assimilated at ECMWF. They should
be regarded as two independent information sources. To clarify this, we added the following
sentence at P11 L17:

**Since VOCALS results were not assimilated in ECMWF analysis, these two data sources can be
regarded as independent.**

Section 4
Page 12. Line 19. "no double or complex structure inside the trapping layer". Please explain
what is being screened out here, and how often it happens.

<Response>

To clarify this condition we added a more complete description as follows at P13 L7:

**Three criteria are utilized for choosing these cases: a spatial distance of less than 300 km, a
temporal difference of less than 3 hours, the lowest height of the GPS-RO refractivity profile
reaches below 1 km to ensure the trapping layer is included. We also exclude the cases with
complex x-h structure inside the trapping layer which can heavily violate the bilinear
assumption, and the cases with multiple ducting layers which makes the equation (9)**

**inapplicable. Approximately 15% of the total number of cases are ruled out by these two additional requirements.**

Figure 12. Suggest rename it Fig. A1, because its only referenced in the appendix.

<Response>

Thank you for the suggestion; the change has been made.

---

## Referee Report (RR1)

Comments on "Correcting negatively biased…" by X. Yu, F. Xie and C. Ao.

The document has been improved since its previous revision. External information can indeed provide relevant data that GNSSRO may not provide if superrefraction is present. This is particularly important for the Planetary Boundary Layer, where this phenomenon is most common. It is interesting to provide examples of such use of external information.

The manuscript is correctly developed and presented. I would only object to the presentation of the following context and implications:

- This call to external information, underscores the fact that GNSSRO is not providing it. The bias properties of the retrieved refractivity profile are no longer those of GNSSRO, but are swapped to those of the external source of data applied to provide the closure (also to some extent those of the algorithm itself) in this case, a PW retrieval.
- I consider particularly important to underscore that some of the properties generally attributed to GNSSRO do not apply to the PBL, not only before the correction presented here, as is well known, but also after, and notably the good traceability. The external source may have its own bias, and the correction of the bias in the low troposphere presented here can only be partial. Although it may help reducing the refractivity bias of the retrieved GNSSRO profile, the retrieved refractivity will inherit the bias of the PW source, which may perhaps be smaller, but not zero, and not necessarily small. Good traceability is achieved with GNSSRO in the upper troposphere, and the low and mid stratosphere. The correction presented here may reduce low tropospheric bias, and this is of interest for certain applications, but cannot make low tropospheric data as traceable as the upper sections of the profiles.

I would thus recommend mentioning this context in the introduction and/or conclusion.

---

## Author Response (AR2)

*Comments on "Correcting negatively biased…" by X. Yu, F. Xie and C. Ao.*

*The document has been improved since its previous revision. External information can indeed provide relevant data that GNSSRO may not provide if superrefraction is present. This is particularly important for the Planetary Boundary Layer, where this phenomenon is most common. It is interesting to provide examples of such use of external information. The manuscript is correctly developed and presented. I would only object to the presentation of the following context and implications:*

- *This call to external information, underscores the fact that GNSSRO is not providing it. The bias properties of the retrieved refractivity profile are no longer those of GNSSRO, but are swapped to those of the external source of data applied to provide the closure (also to some extent those of the algorithm itself) in this case, a PW retrieval.*

- *I consider particularly important to underscore that some of the properties generally attributed to GNSSRO do not apply to the PBL, not only before the correction presented here, as is well known, but also after, and notably the good traceability. The external source may have its own bias, and the correction of the bias in the low troposphere presented here can only be partial. Although it may help reducing the refractivity bias of the retrieved GNSSRO profile, the retrieved refractivity will inherit the bias of the PW source, which may perhaps be smaller, but not zero, and not necessarily small. Good traceability is achieved with GNSSRO in the upper troposphere, and the low and mid stratosphere. The correction presented here may reduce low tropospheric bias, and this is of interest for certain applications, but cannot make low tropospheric data as traceable as the upper sections of the profiles.*

*I would thus recommend mentioning this context in the introduction and/or conclusion.*

**<Response>**

Thank you very much for your suggestion. To clarify we added the following sentences at the end of conclusion:

**It should be recognized that the absolute accuracy of the reconstructed GNSS-RO refractivity will be influenced by the uncertainty of the external constraints. The lower SI traceability of reconstructed refractivity within the PBL compared to the upper troposphere and lower stratosphere (UTLS) region can limit its applicability in long term climate monitoring.**

Also, we would like to kindly remind the reviewer that this paper is submitted by "K.-N. Wang, M. de la Torre Juarez, C. O. Ao, and F. Xie".